# Coral reefs in the Gilbert Islands of Kiribati: Resistance, resilience, and recovery after more than a decade of multiple stressors

**Sara E. Cannon**[1]ᴑ*, **Erietera Aram**[2‡], **Toaea Beiateuea**[2‡], **Aranteiti Kiareti**[2‡], **Max Peter**[2‡], **Simon D. Donner**[1]ᴑ

**1** Department of Geography, University of British Columbia, Vancouver, BC, Canada, **2** Ministry of Fisheries and Marine Resource Development, Coastal Fisheries Division, Bikenibeui, Tarawa, Republic of Kiribati

ᴑ These authors contributed equally to this work.
‡ These authors also contributed equally to this work.
* s.cannon@oceans.ubc.ca

**Data Availability Statement:** The data are held in the public repository Zenodo and the links are as follows: https://doi.org/10.5281/zenodo.4456048 https://doi.org/10.5281/zenodo.4470114.

## Abstract

Coral reefs are increasingly affected by a combination of acute and chronic disturbances from climate change and local stressors. The coral reefs of the Republic of Kiribati's Gilbert Islands are exposed to frequent heat stress caused by central-Pacific type El Niño events, and may provide a glimpse into the future of coral reefs in other parts of the world, where the frequency of heat stress events will likely increase due to climate change. Reefs in the Gilbert Islands experienced a series of acute disturbances over the past fifteen years, including mass coral bleaching in 2004–2005 and 2009–2010, and an outbreak of the corallivorous sea star *Acanthaster* cf *solaris*, or Crown-of-Thorns (CoTs), in 2014. The local chronic pressures including nutrient loading, sedimentation and fishing vary within the island chain, with highest pressures on the reefs in urbanized South Tarawa Atoll. In this study, we examine how recovery from acute disturbances differs across a gradient of human influence in neighboring Tarawa and Abaiang Atolls from 2012 through 2018. Benthic cover and size frequency data suggests that local coral communities have adjusted to the heat stress via shifts in the community composition to more temperature-tolerant taxa and individuals. In densely populated South Tarawa, we document a phase shift to the weedy and less bleaching-sensitive coral *Porites rus*, which accounted for 81% of all coral cover by 2018. By contrast, in less populated Abaiang, coral communities remained comparatively more diverse (with higher percentages of *Pocillopora* and the octocoral *Heliopora*) after the disturbances, but reefs had lower overall hard coral cover (18%) and were dominated by turf algae (41%). The CoTs outbreak caused a decline in the cover and mean size of massive *Porites*, the only taxa that was a 'winner' of the coral bleaching events in Abaiang. Although there are signs of recovery, the long-term trajectory of the benthic communities in Abaiang is not yet clear. We suggest three scenarios: they may remain in their current state (dominated by turf algae), undergo a phase shift to dominance by the macroalgae *Halimeda*, or recover to dominance by thermally tolerant hard coral genera. These findings provide a rare glimpse at the future of coral reefs around the world and the ways they may be affected by climate

**Funding:** This work was supported by a Natural Sciences and Engineering Research Council of Canada Discovery Grant (SDD; www.nserc-crsng.gc.ca, RGPIN-2019-04056). The funders had no role in study design, data collection and analysis, decision to publish, or preparation of the manuscript.

change, which may allow scientists to better predict how other reefs will respond to increasing heat stress events across gradients of local human disturbance.

## Introduction

Phase shifts or regime shifts, changes in the equilibria community in response to a persistent change in environmental conditions [1], are well-documented responses to disturbance on coral reefs. Phase shifts can happen over broad spatial scales, ranging from local (a few kilometers) to regional (thousands of kilometers), and wide time scales (from a period of 1–2 years to decades or longer) [2, 3]. There may also be time lags of several years or more between the disturbance and the resulting change in community composition. Together, these characteristics may make identifying the drivers of phase shifts challenging, but doing so can have important implications for resource management [3].

Once an ecosystem has undergone a phase shift, it is often difficult for it to return to the previous state, particularly when there are multiple disturbances occurring at once. However, there is some evidence that coral reefs that have undergone phase shifts can recover by alleviating the responsible stressors, or restoring the perturbed aspect of the system [1, 4]. One well-known example of this is the case of reefs in Kane'ohe Bay, Hawai'i, where diversion of sewage outflow led to a reversal of a previous phase shift to macroalgae [5]. This example illustrates that identifying and reversing a phase shift requires establishing a link between the drivers and the ecosystem response [3].

On coral reefs, the potential drivers of phase shifts have been well-documented, and may include acute (short-term) disturbances such as climate-driven events (e.g., marine heat waves or tropical cyclones) or chronic (long-term) disturbances (e.g., fishing pressure, nutrient enrichment, or sedimentation, or a combination of these) [6]. The most well-known examples of phase shifts on coral reefs are from coral-dominated to macroalgae-dominated states [7–9], but this may occur more often in the Caribbean than in the Indian and Pacific Oceans [10–12]. Instead, phase shifts to other dominant organisms may be more common. For example, phase shifts to coral taxa with 'weedy' life history strategies [13] have been documented in parts of the Pacific [14], as well as shifts to sponges [15, 16] or corallimorphs [14, 17].

The coral reefs of Tarawa Atoll and its less populated neighbour Abaiang Atoll in the Republic of Kiribati provide a unique opportunity to investigate the role of chronic human disturbances on coral reef recovery from acute disturbances, and could also serve as examples of the ways that reefs in other parts of the world may respond to increasing frequencies of climate-driven heat stress events in the future. These reefs have been exposed to repeated bleaching-level heat stress events in the past 20 years due to the El Niño/Southern Oscillation (ENSO) [18, 19]; during central Pacific El Niño events, slowdown or reversal of easterly trade winds and the South Equatorial Current bring anomalously warm conditions to the central equatorial Pacific [20]. The two atolls, however, experience different levels of local human disturbance. Tarawa is home to roughly 60% of the 110,136 people in Kiribati according to the 2015 census [21]. About 90% of the Tarawa's population is concentrated in communities spread across the southern rim of the atoll (referred to administratively as South Tarawa). By contrast, neighboring Abaiang Atoll, about seven miles north of Tarawa's northern-most point, has less than a tenth of Tarawa's population, about 5,500 people [21]. This difference in human population translates to a difference in chronic human-related pressures like fishing, nutrient loading, and sedimentation. For example, reefs in S. Tarawa experience much higher

fishing pressure than those in N. Tarawa and Abaiang. Although fishers in Abaiang export much of their catch to S. Tarawa, a report from 2004 indicates that fish populations in Abaiang were healthy and showed no signs of overexploitation [22], and research suggests that such small-scale subsistence fisheries are unlikely to substantially affect reef fish assemblages [23].

The gradient in human pressures on the reefs emerged from the colonial history of Kiribati, more so than from recent governance. The Gilbert Islands, known as *Tungaru* by the i-Kiribati prior to colonization, were first settled 2,000–3,000 years ago [24]. The British seized control in 1892, and retained colonial oversight of the Gilberts until Kiribati's independence in 1979, with the exception of a six-year period during and after World War II [24, 25]. During the colonial period, causeways were built that altered natural water flow and sedimentation patterns in S. Tarawa, and to a lesser extent in other atolls like Abaiang, and also blocked fish populations from reaching their traditional spawning and nursery grounds within the lagoon [26–28]. The British centralized and expanded government activity in S. Tarawa, creating a draw for people looking for education, employment, and access to goods and services, and spurring the high population density in Tarawa today [24–26]. The population of S. Tarawa is growing at about 4.5% per year and is expected to double by 2030 [29]. The Kiribati government has attempted to meet the needs of this growing population through major infrastructure projects in S. Tarawa, including many that are underway today [30]. Conversely, there have been few major infrastructure projects in N. Tarawa, Abaiang, and other "outer" atolls, with the exception of causeways [24]. Most outer atolls have experienced steady or declining human populations as people migrate to S. Tarawa [21].

South Tarawa's growing population meant that sewage pollution was also increasing and becoming a growing threat to the health of both people and coral reefs. In 1985, the British completed the first sewage scheme in S. Tarawa [24], which aimed to improve water quality by pumping raw sewage out of three outflows at seven meters depth along the reef crest, via pipes crossing the reef flat. Until recently, the outfalls had not been regularly maintained and leaked untreated sewage onto the reef flats [31]. Notably, these sewage pipes only served a portion of the population on S. Tarawa; as of 2013, about 60% of residents use the ocean, beaches, or lagoon instead of toilets [29].

In 2019, Kiribati's Ministry of Public Works and Utilities completed a project to improve access to toilets (reducing the number of residents not using toilets from 60% to 20%) and to update the sewage system, which included fixing the leaking pipes and moving the outflows from the reef crests to 30m depth [32]. The local government is concerned about the impacts of sewage pollution on reef health [31, 32]. Some coral taxa are unable to tolerate high concentration of nutrients, which can contribute to reef degradation by allowing fleshy macroalgae (as in the case of Kane'ohe Bay, Hawai'i) [5], or weedy coral species like *Porites rus* to outcompete the more sensitive corals [33].

These distinct histories of disturbance have influenced how reefs in Tarawa and Abaiang responded to recent acute disturbances, which include recurrent bleaching events and a CoTs outbreak. The first reported bleaching event at Tarawa and Abaiang occurred in 2004–2005, due to prolonged exposure to higher-than-average SSTs during an El Niño event [34]. There were no reports of bleaching on the outer reefs (in the published literature, grey literature, or via local experts) [34] and cores from massive *Porites* did not provide evidence of past bleaching events [18], although it is possible that mass bleaching occurred during past El Niño events but went unreported. After the first bleaching event, coral genera that are more tolerant of heat stress became more dominant, and researchers observed a similar pattern after a subsequent bleaching event in 2009–2010 [19, 34]. Then, in 2013–2014, an outbreak of the corallivorous Crown-of-Thorns (CoTs) sea star, *Acanthaster* cf *solaris*, occurred in both Tarawa and Abaiang [35]. A CoTs outbreak also occurred in the

1970s [26, 36], and other unreported events may have occurred in the intervening years [28]. Outbreaks of CoTs can cause widespread damage and coral loss on reefs [37], particularly after coral bleaching, when predation may target the thermally-tolerant surviving corals such as massive *Porites* [38, 39].

The weedy coral species *Porites rus* is also thermally tolerant, and previous studies document its spread over time in S. Tarawa, which likely contributed to these highly disturbed reefs' greater resistance to the 2004–2005 and 2009–2010 bleaching events than those experiencing lower human influence [19]. After the first bleaching event ended in 2005, researchers documented a rapid increase in *P. rus* at a single site in Tarawa and hypothesized that it was temporary, but subsequent surveys showed that *P. rus* continued to survive or even proliferate across sites in S. Tarawa despite the subsequent heat stress event [19]. This could suggest that a phase shift was underway among reefs in S. Tarawa, where prior to bleaching in 2004, the coral community was more diverse; S. Tarawa's reefs were home to larger populations of both fast-growing, thermally sensitive genera like *Acropora* and *Pocillopora*, and slower-growing, more thermally tolerant genera like massive *Porites* [19, 34]. In 2012, *P. rus* accounted for the majority of coral cover in S. Tarawa [19], while in Abaiang, coral reefs were still in the process of recovering from the last bleaching event and their future trajectory was unclear [19]. Detecting when a community is in the process of shifting is important because a single perturbation could push that community into a catastrophic shift to a degraded state [40]. If a phase shift has already occurred, identifying it, the parameters that may have caused it, and how the phase shift may affect ecosystem services could inform next steps for management.

Here, we add to the previous research investigating phase shifts on coral reefs by evaluating how benthic communities, including coral, algae, and other key taxa, have responded to multiple stressors in Abaiang and Tarawa Atolls. We examine benthic cover and size frequency data collected in surveys from 2012 through 2018 to evaluate a series of hypotheses about the trajectories of coral reef communities after disturbance. First, we test whether post-bleaching communities shifted towards dominance by disturbance-resistant coral taxa and macroalgae over the study period. Second, we tested whether the shift to *P. rus* in S. Tarawa, documented in previous studies, is persistent and represents a phase shift. Third, we test that the taxa-level response to the CoTs outbreak differs from that of bleaching, with massive *Porites* sensitive to CoTs but more resistant to bleaching. Finally, we examine whether the trajectories of post-bleaching communities differ based on local human disturbance. Our findings provide a rare glimpse at how coral reefs around the world may respond to the increasing frequencies of heat stress events across gradients of local human disturbance and could provide important lessons to guide the future management of coral reef resources in the face of climate change.

## Materials & methods

### Study sites

We sampled 19 sites on the outer reefs across Abaiang and Tarawa Atolls between 2012 and 2018 (Fig 1 and S1 Table). These outer reefs feature spur and groove formations from the reef crest seaward to approximately 10 to 15m depth. The southeastern and eastern reefs are more exposed to prevailing easterly wind directions and swells, and thus have narrower reef terraces than those on the western outer reefs.

Because of the complexity of conducting fieldwork in such a remote location, we were unable to visit a consistent set of sites during each of our visits, resulting in uneven sampling across sites and atolls. Sites were unevenly sampled every two years from 2012 to 2016, and

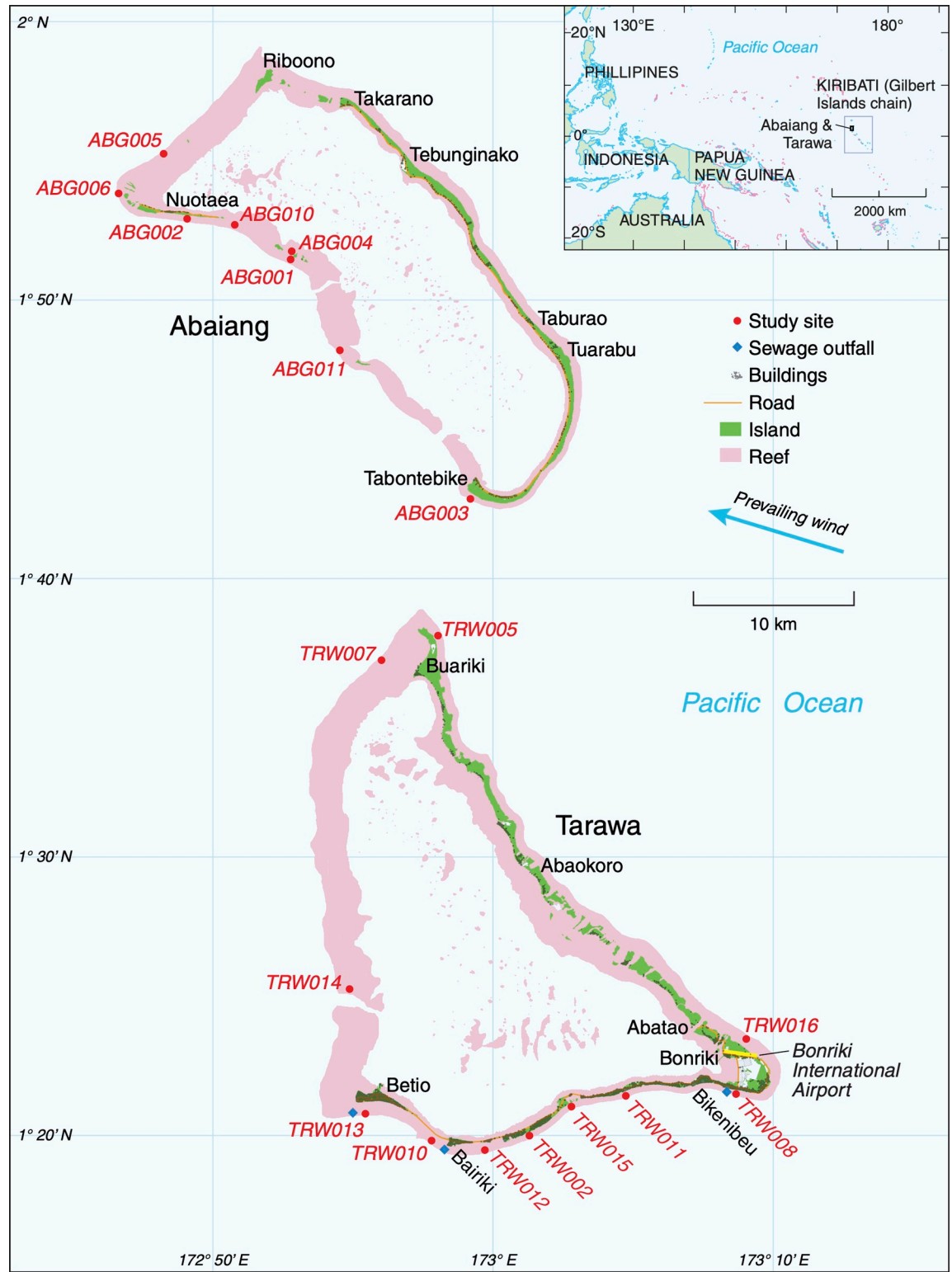

**Fig 1. Study sites in Tarawa and Abaiang.** This figure uses data extracted from the Millennium Coral Reef Mapping Project Version 4.0 [41], and the OpenStreetMap Foundation, available under a CC BY-SA 2.0 license with permission from © OpenStreetMap contributors, original copyright 2012 [42].

were selected to cover a range of habitats, population density, and coastal infrastructure. In 2018, we sampled a wide array of the intended sites, which provides the most recent and complete snapshot of benthic communities in both atolls. We obtained permission to access field sites and conduct scientific research from the Kiribati Ministry of Environment, Lands, and Agricultural Development for the 2018 surveys, and through an established research partnership with the Ministry of Fisheries and Marine Resource Development (MFMRD) for all previous surveys. Repeats of the 2018 surveys planned for 2020 had to be postponed indefinitely because of the COVID-19 pandemic.

All sites are on the ocean side of the atolls (we did not survey sites within the lagoons). Most sites are limited to the south and west rims of each atoll due to unsafe diving conditions and difficulties accessing the northern and northeastern reefs. As in previous work, sites located in the northern tip of North Tarawa (TRW005, TRW007) are grouped with sites from Abaiang because they are physically closer to Abaiang and have similar levels of human disturbance [19, 34]. Going forward, we refer to sites in North Tarawa and Abaiang as 'Abaiang', and sites in South Tarawa as 'Tarawa.'

## Survey methods

All data were collected between April and May in 2012, 2014, 2016, or 2018. Benthic community composition and size-frequency of coral communities were measured using the methods we described in a previous study [12]. A 50-m transect tape was laid randomly at 10-m depth at each site. We took 0.33m²-sized quadrat photos (50.0 cm width by 66.7 cm length) at 50 cm intervals along the transect, for a total of 100 photos per site. These photos were later analyzed to calculate the percent cover of macroalgae and coral genera, with other key benthic taxa, at each site (see Statistical Analysis.).

We also measured the diameter (in cm) of corals in situ along the transect, including all coral colonies $\geq$ 1 cm that lay at least partially within 25-cm on both sides of the tape. We considered corals with separate patches of living tissue > 3-cm apart from each other independent and measured them individually. All corals were identified to the genus level, with the exception of *P. rus*, which we identified to the species level.

All identification relied on taxonomy from Veron [43]. Since this resource was published, the taxonomy of the Favidae family has undergone several changes [44], which we were unable to reflect in these analyses because the size frequency data were collected in situ and the genera thus cannot be corrected to account for the most up-to-date taxonomy. We have included a list of species observed in Tarawa and Abaiang [36] in the supplementary materials for reference (S2 Table).

We used photos from the transects to calculate benthic percent cover using the open-source web tool CoralNet [45], which overlaid 20 random points per photo for 100 photos per site (for a total of 2000 points per site). Each photo covered 0.33 m² (50.0 cm width by 66.7 cm length). We manually identified each point to the genus level for coral and macroalgae, and to functional group for sponges, soft corals, turf algae, crustose coralline algae (CCA), and cyanobacteria. We also identified the coral species *P. rus*, which has a 'weedy' life-history strategy [13], to the species level. To estimate the impacts of 2014's CoTs outbreak, we manually counted the number of recent feeding scars visible in our photo quadrats at the sites we visited that year and identified the genera of the coral with the feeding scars (Fig 2). We considered scars recent if the dead coral patch was still white, and other organisms had not yet colonized the coral skeleton (e.g., algal turf). In this way, we avoided counting scars from bleaching or other causes of mortality, although this method likely underestimates the number of CoTs feeding scars as a result.

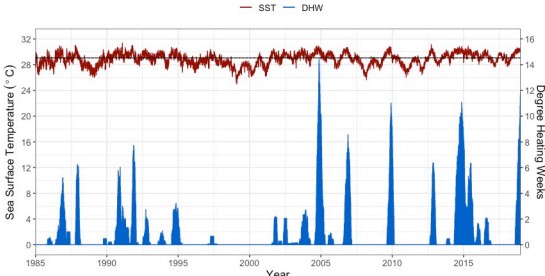

**Fig 2. Daily sea surface temperature (red), Degree Heating Weeks (blue) averaged across study sites from 1985 through 2018, with the maximum monthly mean (black).**

## Human disturbance

We used two different metrics to estimate different aspects of human disturbance. First, we used 2015 census data from Kiribati, which provides the population for each village in the nation [21], to calculate a population metric. Using ArcGIS ArcMap 10.6.1, we measured the distance from each site to the center of the nearest village and then divided the population of that village by the distance. This metric incorporates localised human disturbance that is related to population (such as nutrification and fishing pressure).

We calculated the second human disturbance metric using the Normalized Difference Vegetation Index (NDVI), obtained from the United States Geological Survey's Land Satellite 8 imagery following a method developed in a previous analysis of the neighbouring Republic of the Marshall Islands [12]. NDVI measures the amount of green terrestrial vegetation within a 60-m pixel on a scale of -1.0 to +1.0 and is commonly used to represent the extent of human disturbance on terrestrial ecosystems. This metric captures human alterations of the landscape and is not necessarily affected by local population size. We used NDVI in addition to a population metric to account for land-based disturbances where there are few permanent residents (e.g., Tarawa's Bonriki airport). NDVI has an inverse relationship with disturbance; a high NDVI value indicates a low level of disturbance.

To calculate this metric, we obtained satellite data from November 11, 2017 and February 8, 2018, selected for coverage of all sites and for the low cloud cover on those days. Using ArcGIS ArcMap 10.6.1, we mosaicked the satellite data into a single data layer, and then cast a circle with a 1-km diameter (chosen to minimize overlap of the circles) around each site and traced the landmass that fell within the circle. We then calculated the average NDVI of the landmass, giving us a proxy to rank human influence at each site. For TRW014, the one site that was not within 1-km of land, we used the highest NDVI value from the sites closer to land (indicating the lowest level of disturbance, S1 Table).

## Oceanographic data

Time series of daily SST and Degree Heating Weeks (DHW) for all sites for the years 1985 through 2018 were obtained from 0.05˚ x 0.05˚ resolution CoralTemp SST Version 3.1 satellite-derived data [46]. We calculated the coefficient of variation of SST ($CV_{SST}$) for each site using the entire available dataset of SSTs (1985–2018) to represent temperature variability. We also calculated the monthly maximum mean (MMM), a baseline for estimating heat stress, using a monthly climatology calculated from daily SST values from 1985–1994. Finally, we obtained the satellite-derived monthly chlorophyll-a (chl-a) concentration (in mg m$^3$) via NASA's Moderate Resolution Imaging Spectroradiometer, from July 2002 –May 2019 [47],

and applied a nearest value interpolation in order to fill missing values. We used this full data-set to calculate a mean chl-a value at each site.

## Wind and wave exposure

We created a proxy for wind and wave exposure using the angle of each of our sites to the pre-vailing wind. We first calculated the average prevailing wind direction (p = 111˚, roughly East-Southeast) for 2000–2008 using wind vector data recorded by the Kiribati Meteorological Ser-vice at the station in Betio, Tarawa. We then used Google Earth to measure the compass head-ing perpendicular to the reef crest at each of our sites (C). Using the prevailing wind direction (p) and the compass heading (C), we calculated a normalized exposure metric, where 1 is max-imum exposure and 0 is minimum exposure to the prevailing winds:

$$\text{Exposure Metric} = 1 - (|(p - C)|)/180$$

## Statistical analysis

We investigated change over time at available sites from 2012–2018 to evaluate a series of hypothesis about the trajectories of coral reef communities after disturbance. All statistical analysis was done using R version 4.0.2 [48] and RStudio version 1.3.959 [49]. Plots were cre-ated with the R packages ggplot2 [50] and ggbiplot [51].

We grouped the data into seven categories representing the most ecologically important and prevalent taxa: *Acropora*, *Heliopora*, *Montipora*, *Pocillopora* (genera), Favids (genera of the former family Faviidae), massive *Porites* (morphology of genus *Porites*, including the spe-cies *Porites lutea* and *Porites lobata*), and *P. rus* (species). The octocoral *Heliopora* is included in the coral taxa analysis because of its prevalence throughout the Gilbert Islands. Other key benthic taxa and substrate types analysed include *Halimeda* and *Lobophora* (macroalgae gen-era), crustose coralline algae (CCA), corallimorphs, cyanobacteria, rubble, sand, soft corals, sponges, and turf algae.

We began by testing our first hypothesis that post-disturbance communities were domi-nated by bleaching-resistant coral taxa and macroalgae, and the second hypothesis that the shift in *P. rus* in Tarawa was persistent. To do so, we investigated the change in percent cover for each key taxon and substrate type over time using linear mixed effects models (LMM) with the R package lme4 [52] followed by chi-square tests and Tukey's post hoc tests. The percent cover data met the assumptions of normality and equal variance of resid-uals and were not transformed. The LMM allowed us to account for the uneven sampling of sites across years, by considering 'Site' a random effect and 'Year' a fixed effect. We tested the significance of 'Year' by comparing a null model (without 'Year' as a fixed effect) to a full model (which included the fixed effect) using chi-square tests. Finally, we conducted Tukey's post hoc tests for each of the full LMM using the R package multcomp [53] to inves-tigate the magnitude and direction of change in the percent cover of each taxa between years. We ran the models on three different datasets to ensure that the results were consis-tent despite the uneven sampling of sites: (1) the full dataset containing all sites, (2), from only the sites we visited each year from 2012 through 2018 (ABG001, ABG002, ABG003, TRW002, and TRW010; S1 Table), and finally (3) the sites where we found evidence of CoTs (S1 Table). Because of small sample sizes, we were not able to run the LMM separately for each of the key taxa within the atolls.

To investigate the third hypothesis that massive *Porites* was more sensitive to CoTs but resistant to bleaching, we ran an additional LMM, chi-square test, and Tukey's post hoc tests for massive *Porites* within Abaiang, where we had observed greater prevalence of CoTs scars (discussed further below). We also conducted a similarity percentages analysis (SIMPER)

using the full percent-cover dataset (999 permutations) [54] in the vegan package in R [55], to identify the key taxa driving differences in benthic communities across atolls and years.

We further tested our first hypothesis, along with our third hypothesis about the taxa-level response to the CoTs outbreak, by investigating changes in the size-frequency distributions of key coral taxa over the study period. We did not include *P. rus* in the size frequency analysis, despite its prevalence in Tarawa, because this species grows in extensive mats covering wide areas, and we were unable to distinguish between individual colonies. Size-frequency data were log-transformed to meet assumptions of normality, and critical values for all tests were adjusted using the Bonferroni correction to avoid Type I errors across multiple comparison tests. We first calculated demographic statistics on coral abundance and size for each of the key coral taxa (except for *P. rus)*, including mean size, standard error, skewness and skewness standard error, and kurtosis and kurtosis standard error. We considered skewness and kurtosis values greater than two times the standard error significantly different than normal [56]. We used the Kolmogorov-Smirnov test to compare size frequency distributions across years and atolls [57], and Welch's analysis of variance (ANOVA) tests to examine whether the mean size, coefficient of variation, skewness, or kurtosis for each of the six key taxa (excluding *P. rus*) varied over time [57]. We did not have large enough sample sizes to separate the results of the Welch's ANOVA by atoll because of low sample sizes of some key taxa.

We then used permutational-based multivariate analysis (PERMANOVA), to test our final hypothesis that the trajectories of post-bleaching communities differed by local human disturbance. The analysis, conducted with 99,999 permutations [58], using the vegan package [55], tested for variation in means of all benthic taxa caused by five environmental variables: mean NDVI, the population metric, wind-and-wave exposure, the coefficient of variation of SST, and mean chl-a. To evaluate the impact that time played on the benthic community composition, we also made 'Year' a factor, with each of these environmental factors nested within 'Year.' Although PERMANOVA is not sensitive to collinearity, we excluded atoll as a factor because a factor analysis conducted using the R package psych [59] found that atoll and the population metric were closely correlated (F-statistic = 76.17, p <0.001) and because it did not add to the fit or explanatory power of the model. Instead, in addition to the PERMANOVA that included all sites from both atolls, we also ran the PERMANOVA separately for each atoll to investigate whether there were differences in how each of these factors influenced the benthic compositions within atolls.

## Results

### Disturbance history

We quantified the disturbances affecting reefs during our study period, to investigate our specific hypotheses about coral reef community trajectories post-disturbance. Our analysis of NOAA's Coral Reef Watch historical SST data indicates that reefs in Abaiang and Tarawa experienced bleaching-level heat stress twice between 2012 and 2018. DHWs were greater than 8˚C·week–considered by Coral Reef Watch a Bleaching Alert Level 2 event (severe bleaching and some mortality likely)–in 2012–2013 and 2014–2015, and again six months after the last surveys were conducted in 2018 (Fig 2). No bleaching was observed in our surveys or was reported to the MFMRD research team during our study period. The 2013–14 CoTs outbreak is described in a later section.

We confirmed that the levels of human disturbance on the atolls are statistically distinct based on the population metric and NDVI. Two-way ANOVAs confirmed what we had found when conducting the factor analysis to choose variables for the PERMANOVA (see Statistical Analysis): the atolls differed significantly based on the population metric, and these variables are highly

correlated (F-statistic = 76.17, p < 0.001, adjusted $r^2$ = 0.81). The atolls also varied significantly by NDVI, but the correlation is less strong (F-statistic = 13.304, p < 0.001, adjusted $r^2$ = 0.23).

## Benthic community trajectories between 2012 and 2018

The percent cover data by site and year for each of the key taxa is presented in Table 1. In Abaiang, both the average percent hard coral cover and the average percent macroalgae cover across all sites declined from 2012 to 2016, and then increased from 2016 to 2018 (Fig 3A). In Tarawa, hard coral cover and macroalgae cover followed slightly different trajectories (Fig 3B). Hard coral cover declined from 2012 to 2014, increased from 2014 to 2016, and then remained steady from 2016 to 2018; macroalgae cover declined from 2012 to 2014, and then increased from 2014 to 2018.

The apparent decline in the average live coral cover in Tarawa in 2014 in Table 1 is at least partially a consequence of the sampling methods. The photo quadrat surveys of site TRW010, which features a spur-and-groove system with deep and sometimes wide grooves that consist largely of sand, inadvertently captured more sand than in other years (31.98% of total cover in 2014, compared to 18.1% in 2012 and 17.50% in 2016). In addition, we visited TRW013 and TRW014, which both had lower live coral cover than all other Tarawa sites (12.7% and 3.62%, respectively), for the first time in 2014; this gives a misleading impression that there was a large decline in live coral cover across Tarawa between 2012 and 2014. If we omit these sites and correct the total live coral and *P. rus* cover at TRW010 in 2014 by removing sand from the total percent cover, the decline from 2012 to 2014 is reduced by two-thirds (Fig 3B). Including TRW013 and TRW014 sites in the statistical analyses did not, however, affect the model results, and we therefore present the unadjusted values going forward.

We used LMMs and chi-square tests to investigate our first hypothesis, that post-bleaching communities are dominated by bleaching-resistant coral taxa and macroalgae, and second hypothesis, that the shift to *P. rus* was persistent in Tarawa. The results for the LMMs, including all sites across both atolls, suggest that the percent cover of several key taxa changed significantly from 2012–2018 (Table 2). There was not sufficient data to test whether cover is different between atolls and years. Year was a significant factor driving the change in percent cover of all live coral ($\chi^2$ = 8.36, p = 0.04), as well as changes in Favids ($\chi^2$ = 15.00, p < 0.01), *Heliopora* ($\chi^2$ = 8.59, p = 0.04), *Montipora* ($\chi^2$ = 16.41, p < 0.01), and *Pocillopora* ($\chi^2$ = 14.77, p < 0.01) genera, and massive *Porites* ($\chi^2$ = 9.62, p = 0.02) morphology. Year also was a significant factor in the change in percent cover of all macroalgae genera ($\chi^2$ = 19.68, p < 0.01) and for *Halimeda* specifically ($\chi^2$ = 14.27, p < 0.01), as well as for CCA ($\chi^2$ = 7.95, p = 0.05), rubble ($\chi^2$ = 9.81, p = 0.02), sponges ($\chi^2$ = 14.09, p < 0.01), and turf algae ($\chi^2$ = 14.34, p < 0.01). While we were unable to separate these results by atoll, some of the taxa were only present in one atoll or the other, which allows us to extrapolate which atolls were most affected. For example, *P. rus* and corallimorphs were rare in Abaiang, while massive *Porites* were rare in Tarawa. A repeat of the LMM using only those sites that we visited every year found similar results (S3 Table), so all sites for which we gathered data are used in the following analyses.

The difference between marginal and conditional $r^2$ values in the LMM indicates the extent of the variance explained by the sites surveyed (the random effect) and the year (the fixed effect). For example, the cover of some taxa, such as *Montipora*, were more affected by the year of the survey (conditional $r^2$ = 0.35, or 35% of the variance) than the sites surveyed (marginal r2—conditional $r^2$ = 0.10, or 10%) (Table 2). For all live coral cover, the entire model explained 89% of the variance, with the majority of that (86%) explained by differences across sites, and only 3% explained by the difference across years. Similarly, the entire model explained most of the variance in massive *Porites* (79%), with sites surveyed explaining 72% of the variance and

**Table 1. Mean and standard deviations of the percent cover of key benthic category, for each survey year by atoll.**

| Category | Atoll | 2012 | 2014 | 2016 | 2018 |
|---|---|---|---|---|---|
| **Hard Coral Taxa** | | | | | |
| All Live Coral | Abaiang | 20.97 ± 5.68 | 16.37 ± 7.75 | 11.47 ± 6.86 | 18.26 ± 7.45 |
| | Tarawa | 39.28 ± 10.60 | 14.16 ± 9.83 | 27.84 ± 19.17 | 28.26 ± 16.87 |
| *Acropora* | Abaiang | 0.20 ± 0.20 | 0.17 ± 0.17 | 0.02 ± 0.04 | 0.21 ± 0.13 |
| | Tarawa | 0.33 ± 0.41 | 1.09 ± 1.26 | 0.09 ± 0.04 | 0.11 ± 0.13 |
| Favids | Abaiang | 2.41 ± 0.57 | 0.28 ± 0.27 | 0.34 ± 0.46 | 2.35 ± 1.25 |
| | Tarawa | 0.41 ± 0.48 | 0.24 ± 0.40 | 0.07 ± 0.06 | 0.99 ± 1.24 |
| *Heliopora* | Abaiang | 6.00 ± 4.25 | 8.59 ± 5.28 | 3.03 ± 2.31 | 6.52 ± 4.40 |
| | Tarawa | 3.89 ± 3.09 | 2.40 ± 1.50 | 1.91 ± 1.43 | 1.81 ± 1.56 |
| *Montipora* | Abaiang | 0.33 ± 0.01 | 0.09 ± 0.11 | 0.00 | 0.58 ± 0.27 |
| | Tarawa | 0.18 ± 0.17 | 0.23 ± 0.26 | 0.08 ± 0.09 | 0.37 ± 0.42 |
| *Pocillopora* | Abaiang | 1.20 ± 0.21 | 1.09 ± 0.88 | 0.57 ± 0.33 | 2.90 ± 1.65 |
| | Tarawa | 1.56 ± 1.14 | 1.04 ± 1.10 | 1.29 ± 1.14 | 2.07 ± 1.14 |
| Massive *Porites* | Abaiang | 6.17 ± 1.98 | 4.92 ± 4.93 | 3.09 ± 4.85 | 3.23 ± 2.25 |
| | Tarawa | 0.71 ± 0.26 | 0.94 ± 1.20 | 0.23 ± 0.16 | 0.65 ± 0.72 |
| *P. rus* | Abaiang | 0.11 ± 0.03 | 0.06 ± 0.05 | 0.26 ± 0.42 | 0.07 ± 0.04 |
| | Tarawa | 28.31 ± 13.97 | 5.05 ± 4.19 | 22.19 ± 19.83 | 23.49 ± 19.84 |
| **Macroalgae Taxa** | | | | | |
| All Macroalgae | Abaiang | 38.13 ± 12.09 | 13.86 ± 5.96 | 7.93 ± 7.19 | 13.29 ± 8.70 |
| | Tarawa | 9.90 ± 6.21 | 2.61 ± 3.88 | 5.40 ± 4.91 | 7.99 ± 3.96 |
| *Halimeda* | Abaiang | 38.10 ± 12.04 | 13.74 ± 5.92 | 2.68 ± 0.40 | 13.04 ± 8.71 |
| | Tarawa | 2.40 ± 3.47 | 2.27 ± 4.30 | 1.88 ± 3.44 | 3.09 ± 4.03 |
| *Lobophora* | Abaiang | 0.00 | 0.10 ± 0.14 | 5.17 ± 6.80 | 0.12 ± 0.11 |
| | Tarawa | 8.00 ± 4.06 | 0.86 ± 1.37 | 4.21 ± 5.06 | 6.70 ± 4.64 |
| **Other Benthic Taxa** | | | | | |
| CCA | Abaiang | 6.99 ± 0.52 | 10.38 ± 3.31 | 12.27 ± 2.31 | 9.83 ± 4.72 |
| | Tarawa | 4.48 ± 2.04 | 9.34 ± 4.95 | 9.54 ± 4.42 | 9.02 ± 3.74 |
| Corallimorphs | Abaiang | 0.00 | 0.05 ± 0.00 | 0.00 | 0.07 ± 0.03 |
| | Tarawa | 8.85 ± 15.12 | 2.79 ± 2.38 | 2.64 ± 4.32 | 4.30 ± 5.74 |
| Cyanobacteria | Abaiang | 2.76 ± 1.87 | 2.33 ± 0.66 | 1.96 ± 1.67 | 2.43 ± 2.42 |
| | Tarawa | 4.50 ± 1.65 | 3.09 ± 5.22 | 4.69 ± 4.43 | 8.67 ± 4.86 |
| Rubble | Abaiang | 2.49 ± 2.07 | 6.43 ± 4.45 | 8.90 ± 5.30 | 4.32 ± 3.10 |
| | Tarawa | 2.07 ± 1.20 | 7.69 ± 4.45 | 3.29 ± 3.46 | 2.67 ± 3.02 |
| Sand | Abaiang | 4.06 ± 3.21 | 6.70 ± 5.07 | 4.65 ± 2.85 | 8.31 ± 5.80 |
| | Tarawa | 9.48 ± 12.20 | 17.46 ± 14.19 | 8.74 ± 9.25 | 5.20 ± 7.78 |
| Soft Coral | Abaiang | 0.02 ± 0.04 | 0.27 ± 0.25 | 0.25 ± 0.44 | 0.13 ± 0.16 |
| | Tarawa | 0.13 ± 0.19 | 0.00 | 0.04 ± 0.05 | 0.14 ± 0.12 |
| Sponges | Abaiang | 1.95 ± 0.41 | 1.34 ± 0.48 | 3.59 ± 1.19 | 2.05 ± 0.91 |
| | Tarawa | 6.59 ± 3.33 | 1.58 ± 1.03 | 4.05 ± 1.69 | 4.52 ± 1.76 |
| Turf Algae | Abaiang | 22.45 ± 9.92 | 42.25 ± 4.25 | 48.73 ± 3.76 | 41.22 ± 6.96 |
| | Tarawa | 21.58 ± 13.28 | 41.49 ± 13.61 | 35.52 ± 24.98 | 31.69 ± 13.71 |

the year of the survey explain 7% (Table 2). This is not surprising, given that we found most massive *Porites* colonies at sites in Abaiang.

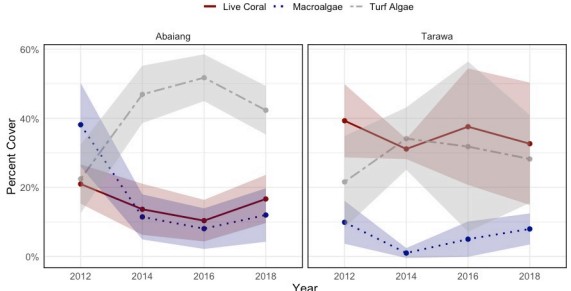

**Fig 3. Time series of mean percent cover of live coral, macroalgae, and turf algae by atoll.** The shaded area around each line represents the standard deviation of the percent cover. The live coral excludes TRW013 or TRW014, which we did not visit in 2012.

We followed the LMM with Tukey contrasts for multiple comparison of means, to produce specific quantitative estimates of how the percent cover of each taxa changed over time across all sites and years (significant results in Table 3; full results in S4 Table). The mean live coral declined by an estimated 7.59% across all sites from 2012 to 2014 (z = -2.74, p = 0.03). The percent cover of Favids, *Montipora*, and *Pocillopora* all increased modestly between 2014 and 2018 and between 2016 and 2018 (Table 3), while massive *Porites* was the only key coral taxa

**Table 2. Results of linear mixed effects models for each key benthic category.** Statistically significant results at α = 0.05 are in bold, while those that are significant at α = 0.10 are underlined.

| Categories | $\chi^2$ | p | Marg $R^2$ | Cond $R^2$ |
|---|---|---|---|---|
| **Hard Coral Taxa** | | | | |
| All Live Coral | **8.36** | **0.04** | **0.03** | **0.89** |
| *Acropora* | 7.74 | 0.05 | 0.18 | 0.19 |
| Favids | **15.00** | **<0.01** | **0.32** | **0.49** |
| *Heliopora* | **8.59** | **0.04** | **0.05** | **0.79** |
| *Montipora* | **16.41** | **<0.01** | **0.35** | **0.45** |
| *Pocillopora* | **14.77** | **<0.01** | **0.14** | **0.68** |
| *Porites* (Massive) | **9.62** | **0.02** | **0.07** | **0.79** |
| *P. rus* | 6.75 | 0.08 | 0.02 | 0.91 |
| **Macroalgae Taxa** | | | | |
| All Macroalgae | **19.68** | **<0.01** | **0.29** | **0.53** |
| *Halimeda* | **14.27** | **<0.01** | **0.26** | **0.55** |
| *Lobophora* | 5.15 | 0.16 | 0.09 | 0.33 |
| **Other Benthic Categories** | | | | |
| CCA* | **7.95** | **0.05** | **0.14** | **0.38** |
| Corallimorphs | 4.06 | 0.26 | 0.12 | 0.51 |
| Cyanobacteria | 5.21 | 0.16 | 0.08 | 0.41 |
| Rubble | **9.81** | **0.02** | **0.14** | **0.53** |
| Sand | 6.33 | 0.10 | 0.06 | 0.71 |
| Soft Coral | 1.48 | 0.69 | 0.04 | 0.40 |
| Sponges | **14.09** | **<0.01** | **0.16** | **0.64** |
| Turf algae | **14.34** | **<0.01** | **0.17** | **0.66** |

*CCA = crustose coralline algae.

**Table 3. Significant results of Tukey contrasts multiple comparisons of means for changes in percent cover, using α = 0.05.**

| Categories | Years | Estimate | St. Error | z-value | p-value |
|---|---|---|---|---|---|
| *Hard Coral Taxa* | | | | | |
| All Live Coral | 2012–2014 | -7.59 | 2.77 | -2.74 | 0.03 |
| Favids | 2014–2018 | 1.40 | 0.42 | 3.32 | <0.01 |
| | 2016–2018 | 1.40 | 0.42 | 3.38 | <0.01 |
| Montipora | 2014–2018 | 0.30 | 0.11 | 2.70 | 0.03 |
| | 2016–2018 | 0.42 | 0.10 | 4.18 | <0.01 |
| Pocillopora | 2014–2018 | 1.06 | 0.33 | 3.25 | <0.01 |
| | 2016–2018 | 1.05 | 0.31 | 3.43 | <0.01 |
| Massive Porites | 2012–2016 | -1.92 | 0.64 | -3.00 | 0.01 |
| *Macroalgae Taxa* | | | | | |
| All Macroalgae | 2012–2014 | -14.70 | 3.47 | -4.24 | <0.01 |
| | 2012–2016 | -14.46 | 3.42 | -4.23 | <0.01 |
| | 2012–2018 | -11.61 | 3.26 | -3.57 | <0.01 |
| Halimeda | 2012–2014 | -13.99 | 4.47 | -3.13 | <0.01 |
| | 2012–2016 | -19.13 | 4.61 | -4.15 | <0.01 |
| | 2012–2018 | -12.95 | 4.44 | -2.92 | 0.02 |
| *Other Benthic Categories* | | | | | |
| CCA | 2012–2016 | 4.53 | 1.65 | 2.74 | 0.03 |
| Rubble | 2012–2014 | 3.92 | 1.41 | 2.79 | 0.03 |
| | 2014–2018 | -3.02 | 1.16 | -2.60 | 0.04 |
| Sponges | 2012–2014 | -2.41 | 0.71 | -3.40 | <0.01 |
| | 2014–2016 | 1.96 | 0.61 | 3.19 | <0.01 |
| | 2014–2018 | 1.70 | 0.59 | 2.89 | 0.02 |
| Turf Algae | 2012–2014 | 16.88 | 5.04 | 3.35 | <0.01 |
| | 2012–2016 | 18.67 | 4.85 | 3.89 | <0.01 |
| | 2012–2018 | 13.38 | 4.61 | 2.90 | 0.02 |

to decline significantly at α = 0.05 (by 1.92% between 2012–2016, z = -3.00, p = 0.01), although the decline between 2012 and 2018 was significant at α = 0.10 (S4 Table). While the percent change of the total benthic taxa across both atolls was small, massive *Porites* in Abaiang declined in half (from 6.17 ± 1.98% in 2012 to 3.09 ± 4.85% in 2016, Table 1). The Tukey results show that macroalgae declined significantly over time, while turf algae increased during the same periods. In Abaiang, there was roughly a two-thirds decline of *Halimeda* over our study period (from 38.10 ± 12.04% in 2012 to 13.04 ± 8.71% in 2018, Table 1), and an almost doubling of turf algae (from 22.45 ± 9.92 in 2012 to 41.22 ± 6.96 in 2018, Table 1). All other key taxa changed by less than 5% over the given time periods (Table 3).

In addition to changes in the percent cover of these key taxa, we found that the average size (in cm) of key coral taxa declined between 2012 and 2018 (we did not include *P. rus* in this analysis because it is difficult to distinguish individual colonies). The only exception is *Acropora* in Tarawa, which increased from a mean size of 10.3 cm in 2012 to 12.3 cm in 2018 (Table 4) but remained rare (average of two colonies per site in 2018). With some exceptions (Favids and *Acropora* in Tarawa), skewness increased for most of the key taxa, showing an overall shift to smaller sizes. Kurtosis, a measure of the steepness of the size-distribution curve, also increased for most coral taxa, suggesting that the size-frequency of each taxa has become more concentrated among a smaller range of values. By 2018, the size of most key coral taxa

**Table 4. Size-frequency statistics for key coral taxa.** Skewness and kurtosis values that are significantly different from normal (greater than two times the standard error) are in bold.

| | Year | Abaiang | | | | | | | Tarawa | | | | | | |
|---|---|---|---|---|---|---|---|---|---|---|---|---|---|---|---|
| | | n* | Mean Size (cm) | Standard Error (SE) | Skewness | Skewness SE | Kurtosis | Kurtosis SE | n* | Mean Size (cm) | Standard Error (SE) | Skewness | Skewness SE | Kurtosis | Kurtosis SE |
| *Acropora* | 2012 | 3 | 22.0 | 4.8 | 1.0 | 1.5 | -0.1 | 3.1 | 4 | 10.3 | 1.5 | 1.6 | 1.3 | 2.4 | 2.5 |
| | 2014 | 1 | 24.3 | 7.5 | -1.8 | 2.4 | 3.6 | 4.9 | 10 | 15.0 | 6.3 | **6.0** | **0.8** | **36.4** | **1.6** |
| | 2016 | 1 | 5.0 | 0.0 | NA | 2.8 | NA | 5.7 | 1 | 9.9 | 2.2 | 0.5 | 1.9 | -1.3 | 3.7 |
| | 2018 | 6 | 12.9 | 0.9 | 1.4 | 0.7 | 2.2 | 1.5 | 2 | 12.3 | 1.9 | 1.6 | 1.3 | 2.4 | 2.6 |
| *Faviids* | 2012 | 22 | 22.6 | 1.7 | 1.0 | 0.6 | 0.1 | 1.2 | 7 | 10.5 | 1.4 | 1.7 | 0.9 | 3.2 | 1.9 |
| | 2014 | 10 | 5.2 | 0.3 | 1.4 | 0.6 | 2.0 | 1.1 | 19 | 9.0 | 0.8 | 1.5 | 0.6 | 2.8 | 1.1 |
| | 2016 | 11 | 6.8 | 0.4 | 1.2 | 0.8 | 0.9 | 1.5 | 10 | 5.1 | 0.3 | 1.7 | 0.6 | 3.2 | 1.2 |
| | 2018 | 142 | 8.2 | 0.1 | **4.7** | **0.1** | **48.5** | **0.3** | 16 | 8.8 | 0.4 | 1.3 | 0.4 | 1.9 | 0.8 |
| *Heliopora* | 2012 | 16 | 56.1 | 5.4 | 1.4 | 0.7 | 2.5 | 1.4 | 15 | 22.6 | 2.9 | 2.9 | 0.6 | **9.9** | **1.3** |
| | 2014 | 18 | 22.0 | 1.5 | 1.6 | 0.4 | 2.3 | 0.8 | 18 | 15.4 | 1.2 | 0.9 | 0.6 | 0.2 | 1.2 |
| | 2016 | 7 | 10.3 | 1.4 | 1.6 | 0.9 | 1.7 | 1.8 | 43 | 9.7 | 0.8 | **4.9** | **0.3** | **30.6** | **0.6** |
| | 2018 | 69 | 13.5 | 0.7 | **4.1** | **0.2** | **23.9** | **0.4** | 32 | 10.2 | 0.6 | **3.6** | **0.3** | **18.0** | **0.6** |
| *Montipora* | 2012 | 1 | 18.3 | 8.8 | 0.9 | 2.8 | NA | 5.7 | 2 | 28.5 | 5.3 | 0.8 | 2.0 | 0.3 | 4.0 |
| | 2014 | 2 | 6.7 | 1.4 | 2.0 | 1.4 | 4.4 | 2.8 | 2 | 22.8 | 5.7 | 0.4 | 2.0 | 0.9 | 4.0 |
| | 2016 | 3 | 8.8 | 0.8 | 0.8 | 1.5 | 1.9 | 3.0 | 1 | 7.4 | 0.9 | -1.3 | 1.9 | 1.9 | 3.7 |
| | 2018 | 17 | 12.9 | 0.6 | 1.6 | 0.4 | 3.8 | 0.8 | 2 | 16.0 | 2.1 | 1.0 | 1.3 | 0.8 | 2.5 |
| *Pocillopora* | 2012 | 11 | 25.4 | 2.0 | 0.3 | 0.8 | -0.3 | 1.7 | 14 | 21.0 | 1.5 | 1.0 | 0.6 | 0.5 | 1.3 |
| | 2014 | 3 | 21.9 | 1.9 | -0.6 | 1.0 | -0.3 | 2.0 | 9 | 21.4 | 2.1 | 1.1 | 0.8 | 3.2 | 1.7 |
| | 2016 | 4 | 16.3 | 3.5 | 1.2 | 1.3 | 0.5 | 2.5 | 13 | 21.9 | 1.3 | 0.8 | 0.6 | 1.6 | 1.1 |
| | 2018 | 69 | 10.2 | 0.4 | 2.3 | 0.2 | 5.4 | 0.4 | 121 | 16.1 | 1.0 | 1.9 | 0.4 | 5.4 | 0.7 |
| Massive *Porites* | 2012 | 18 | 45.5 | 4.9 | 1.8 | 0.7 | 3.7 | 1.3 | 2 | 31.7 | 7.5 | 0.5 | 2.0 | -0.6 | 4.0 |
| | 2014 | 16 | 20.7 | 2.0 | 2.0 | 0.4 | 3.5 | 0.9 | 2 | 15.9 | 6.1 | 2.1 | 1.6 | 4.6 | 3.3 |
| | 2016 | 5 | 8.1 | 0.9 | 1.8 | 1.2 | 4.8 | 2.3 | 5 | 6.5 | 0.5 | 0.2 | 0.9 | -0.9 | 1.9 |
| | 2018 | 72 | 10.7 | 0.5 | **5.5** | **0.2** | **40.6** | **0.4** | 4 | 19.1 | 3.2 | 2.1 | 0.9 | 5.5 | 1.7 |

n* = n-values normalized to the number of sites we visited each year in each atoll.

decreased, the range of sizes also decreased, and smaller corals dominated most of the benthic communities in each atoll, compared to 2012. None of the changes in mean size (in cm) and skewness among the key taxa between 2012 and 2018 were significant according to Welch's two-way ANOVA tests, although the kurtosis and coefficient of variation did change significantly for *Montipora* and massive *Porites* (S5 Table). However, we were unable to separate this analysis by atoll because of the low sample sizes for some of the key taxa within certain years, and some of our sample sizes were too small to test (e.g., for *Acropora*).

Two-sample Kolmogorov-Smirnov (KS) tests found that the size distribution of Faviids (D = 0.82, p-adjusted < 0.01), *Heliopora* genus (D = 0.51, p-adjusted <0.01), and the massive morphology of the *Porites* genus (D = 0.53, p-adjusted < 0.01) had significantly changed in Abaiang, but not Tarawa, between all years of the dataset. The results are similar comparing 2012 and 2018, with Faviids (D = 0.69, p-adjusted < 0.01) and massive *Porites* (D = 0.76, p-adjusted < 0.01), which are uncommon in Tarawa, size distributions differing only in Abaiang.

Finally, we used the SIMPER analysis to identify and rank the taxa that had contributed the most to changes in the percent cover over time. Because atoll was closely correlated to the population metric, we ran the analysis separately for each atoll. In Abaiang, the changes in percent cover of *Halimeda* (macroalgae) and turf algae were significant components of the variation of benthic taxa (Table 5). In Tarawa, *P. rus*, turf algae, corallimorphs, sand, and *Lobophora* (macroalgae) were the most influential taxa to change, although none of these taxa were statistically significant components of variation across time.

At the end of our study period, sites in Tarawa had higher percent cover of live coral, despite experiencing greater localized human disturbance (Fig 4A). However, the live coral cover in Tarawa was almost entirely composed of the weedy coral *P. rus* (81.48% of all coral cover) while sites in Abaiang were composed of comparatively more diverse coral assemblages, including higher cover of the *Pocillopora* and the octocoral *Heliopora* (Fig 4B).

### The effect of the CoTS outbreak

To test the hypothesis that the taxa-level response to the CoTs outbreak differs from that of bleaching, we counted the number of recent CoTs scars by taxa at all sites in 2014. Almost all the sites where CoTs scars were observed were in Abaiang; only one (TRW010) was in Tarawa (S1 Table). CoTs scars were most frequently noted on massive *Porites*, which accounted for 130 of the 146 observed scars (89%). The remaining 11% of scars were observed on the species *P. rus* (n = 6), the Favid family (n = 4), and the genera *Platygyra* (n = 2) and *Turbinaria* (n = 2), the table morphology of the *Acropora* genus (n = 1), and the encrusting morphology of the genus *Porites* (n = 1).

We also conducted a LMM using only those sites with evidence of CoTs but found similar results to LMM using all sites (S3 Table). While we did not have enough data to test how the percent cover of each of the key categories had changed across years within each atoll, we were able to use a LMM to specifically investigate how massive *Porites* had changed across years within Abaiang only. We found that the year of the survey was a significant factor in the model ($\chi^2$ = 8.27, p = 0.04). The Tukey analysis also showed that within Abaiang, massive *Porites* declined significantly between 2012 and 2016, by 3.00% (St. Err = 1.11, z = -2.72, p = 0.03), as suggested by CoTs scars observational data.

The cover of massive *Porites* declined by over 20% between 2012 and 2014 in Abaiang (from 6.17% cover in 2012 to 4.92% in 2014) and by 50% between 2012 and 2016 (3.09% in

**Table 5. Most influential taxa in benthic community difference between 2012–2018, as identified by SIMPER analysis.** Statistically significant results are in bold.

| 2012–2018 | | | | | |
|---|---|---|---|---|---|
| **Abaiang** | | | **Tarawa** | | |
| Taxa | %* | p | Taxa | %* | p |
| *Halimeda* | **33.38** | **0.01** | *P. rus* | 23.50 | 0.61 |
| **Turf algae** | **24.97** | **0.02** | Turf algae | 19.55 | 0.90 |
| Sand | 8.01 | 0.23 | Corallimorphs | 0.14 | 0.10 |
| CCA** | 4.84 | 0.36 | Sand | 8.26 | 0.97 |
| -- | -- | -- | *Lobophora* | 6.16 | 0.52 |
| -- | -- | -- | Cyanobacteria | 6.08 | 0.66 |
| **Total (%)** | **72.21** | -- | **Total (%)** | **72.69** | -- |

* Percent contribution.

** Crustose coralline algae.

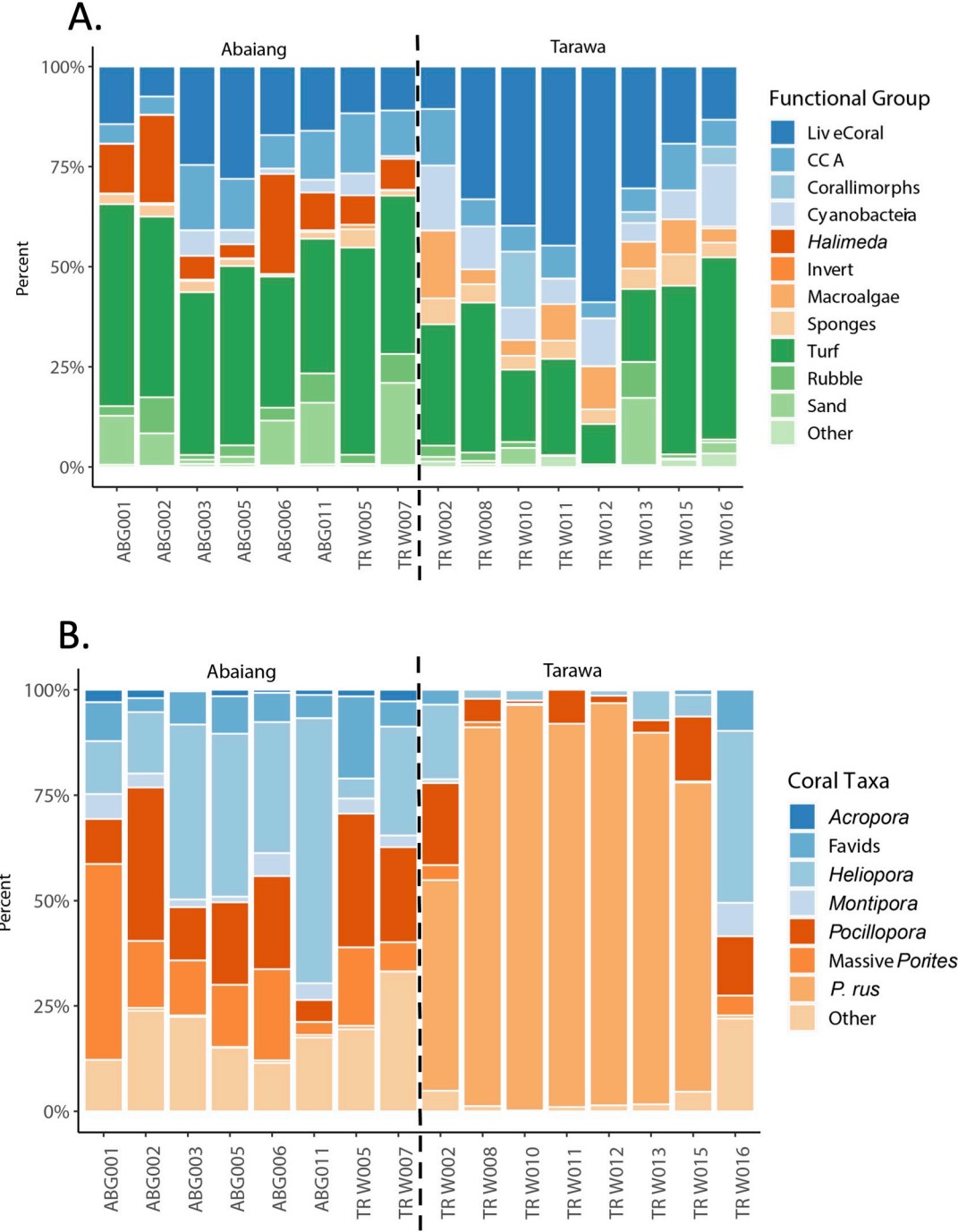

**Fig 4. Percent cover of benthic taxa in 2018.** (A) Percent cover of key functional groups. (B) Percent cover of key reef-building coral taxa.

2016), and the latter decline was statistically significant (Tukey post-hoc test, z = -2.72, p = 0.03). The average size of massive *Porites* also declined in Abaiang, from 45.5 cm in 2012 to 10.7 cm in 2018. While this change was not statistically significant, the KS test confirms that the size distribution of massive *Porites* shifted significantly over time in Abaiang (S3 Table).

## The role of human disturbance

We used a PERMANOVA to investigate our final hypothesis, that local human disturbance was a significant factor driving changes in community composition over time (Table 6). Overall, the model containing data from all sites accounted for 44.74% of the variation in the benthic communities (Pseudo-F = 4.86, p-value < 0.01). The model results show that time alone ('year') accounted for little of the variation in percent cover across the dataset (3% of the variation) and was not statistically significant (Pseudo-F = 1.74, p-value = 0.14). The interaction between year and mean NDVI contributed most to the variation in the percent cover, and collectively explained 25% of the variation in community composition across sites. The other statistically significant interactions were with $CV_{SST}$ (7% of the variance) and the population metric explained (6% of the variance).

We also ran the PERMANOVA separately for each atoll (Table 6). In Abaiang, the full model accounted for 51.14% of the variation in benthic community composition across sites (Pseudo-F = 2.27, p = 0.01), while in Tarawa it accounted for 62.87% of the variation (Pseudo-F = 4.52, p < 0.01). In Abaiang, 'Year' was the only significant factor in the model at α = 0.05 (F = 3.53, p = 0.02), explaining 13% of the difference in benthic composition across sites. The interactions between Year and Mean NDVI (Pseudo-F = 2.67, p = 0.05), Year and $CV_{SST}$ (Pseudo-F = 2.25, p = 0.08), and Year and Exposure (Pseudo-F = 2.62, p = 0.05) were significant at α = 0.10 (accounting for 10%, 8%, and 10% of the variation in the benthos across sites, respectively). By contrast, in Tarawa, the percent cover changed differently across years based on the mean NDVI (Pseudo-F = 10.55, p < 0.01) and the population metric (Pseudo-F = 3.21, p = 0.03). The interactions between Year and the two metrics of human influence collectively accounted for 31% of the variation across sites in Tarawa (Year and Mean NDVI accounted for 24% of the variation, and Year and the population metric accounted for 7%). The interaction between Year and Chl-a was also significant in Tarawa (Pseudo-F = 7.99, p < 0.01), accounting for 19% of the variation across sites.

## Discussion

This study investigated how the benthic communities at sites experiencing different levels of localized human-related degradation responded to a series of acute environmental disturbances, and also identified and documented the coral reef community trajectories after those disturbances. In addition to the coral bleaching events in the Gilberts that preceded our study period (2004–2005 and 2009–2010), which were described and evaluated in detail by Donner and Carilli (2019), there was a CoTs outbreak in 2014 and bleaching-level heat stress from

**Table 6. PERMANOVA of predictors of benthic composition across sites.** Statistically significant values are in bold.

| | All Sites | | | | Abaiang | | | | Tarawa | | | |
|---|---|---|---|---|---|---|---|---|---|---|---|---|
| Factor | SS[1] | R[2] | F | p | SS[1] | R[2] | F | p | SS[1] | R[2] | F | p |
| Year | 0.14 | 0.03 | 1.74 | 0.14 | **0.14** | **0.13** | **3.52** | **0.02** | 0.10 | 0.04 | 1.82 | 0.14 |
| Year: Mean NDVI | **1.27** | **0.25** | **16.16** | **<0.01** | 0.11 | 0.10 | 2.67 | 0.05 | **0.58** | **0.24** | **10.55** | **<0.01** |
| Year: Mean Chl-a | 0.11 | 0.02 | 1.43 | 0.21 | 0.05 | 0.05 | 1.27 | 0.26 | **0.44** | **0.19** | **7.99** | **<0.01** |
| Year: $CV_{SST}$ | **0.37** | **0.07** | **4.71** | **<0.01** | 0.09 | 0.08 | 2.25 | 0.08 | 0.09 | 0.04 | 1.65 | 0.18 |
| Year: Population Metric | **0.32** | **0.06** | **4.11** | **<0.01** | 0.05 | 0.05 | 1.27 | 0.26 | **0.18** | **0.07** | **3.21** | **0.03** |
| Year: Exposure | 0.08 | 0.02 | 1.01 | 0.37 | 0.10 | 0.10 | 2.62 | 0.05 | 0.10 | 0.04 | 1.87 | 0.13 |
| Residual | 2.83 | 0.55 | -- | -- | 0.52 | 0.49 | -- | -- | 0.88 | 0.37 | -- | -- |

[1]Sum of Squares.

2014 through 2016, although the MFMRD did not report any bleaching and evidence was not apparent in our data. While it is possible that bleaching occurred between 2014 and 2016 and went unobserved, corals may not have bleached because the community composition in both atolls has shifted towards dominance by more heat-resistant coral taxa, such as *P. rus* in Tarawa and *Heliopora* and massive *Porites* in Abaiang [19]. Also, while the heat stress in 2014 through 2016 was long-lasting, the magnitude never reached the levels of the 2004–05 and 2009–10 events (>12˚C·week) (Fig 2).

The analysis broadly confirmed our four hypotheses about the trajectories of these post-disturbance coral reef communities. First, the LMM analyses suggest that post-bleaching communities shifted towards dominance by disturbance-resistant taxa over the study period. Second, those results also suggest the shift to *P. rus* in Tarawa, documented in previous studies, were persistent and could represent a phase shift. Third, the scar observations and coral cover data indicate that the taxa-level response to the CoTs outbreak differed from that of bleaching (with massive *Porites* more sensitive to CoTs but resistant to bleaching). Finally, the analysis of the 2018 data indicates that the benthic communities after the sequence of acute disturbances differed based on the level of local human disturbance.

Based on these hypotheses, we propose that most sites in Tarawa and Abaiang have followed the trajectories shown in Fig 5, and we refer to these trajectories to guide our discussion of the results. As indicated by the PERMANOVA, these distinct trajectories were driven by different levels of human disturbance across Tarawa and Abaiang. Before the bleaching event in 2004–2005, previous studies reported that outer reefs in Abaiang had high (>50%) coral cover,

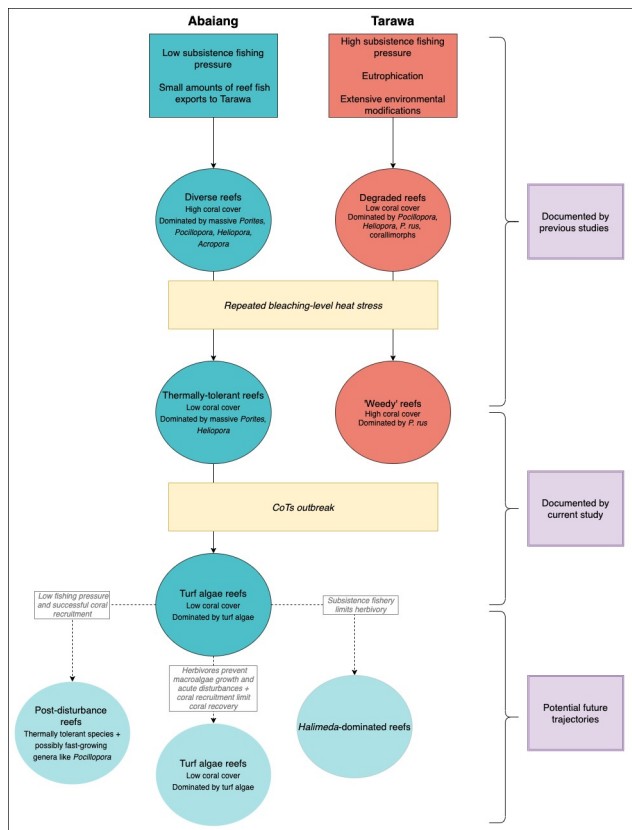

**Fig 5. Proposed drivers of community composition on reefs in Tarawa and Abaiang.**

and communities were dominated by massive *Porites*, *Pocillopora*, *Heliopora*, Favids, the macroalgae *Halimeda*, and to a lesser extent, *Acropora*. In Tarawa, outer reef communities tended to have low coral cover, and were dominated by *Pocillopora*, *P. rus*, and *Heliopora* [34, 36]. After the 2004–2005 and 2009–2010 bleaching events, reefs in Abaiang shifted to lower coral cover dominated by hardier, slower-growing corals (like massive *Porites*), and *Heliopora*, a branching thermally-tolerant octocoral, while reefs in Tarawa adjacent to growing human communities shifted to higher coral cover that was almost entirely composed of *P. rus* [19, 34]. While Abaiang's reefs were more impacted by bleaching and CoTs, they may still be able to recover to a similar coral-dominated state as what was there prior to the acute disturbances, due to the relative lack of localized human influence. Sites in Tarawa, by contrast, appear to have undergone a phase shift and settled into their current state, with sites that are dominated by *P. rus*. Below we discuss the trajectories of the coral reefs in Tarawa and Abaiang, respectively, the impact of CoTs relative to bleaching, and finally the implications of these results for future resource management in the Gilbert Islands.

## Tarawa

The results of this and past studies support our first and second hypotheses, that most outer reefs in Tarawa adjacent to dense human communities have shifted to dominance by disturbance-tolerant taxa, and that the coral reef communities adjacent to dense human populations in S. Tarawa have undergone a phase shift to communities dominated by a single coral species, *P. rus*. Here, when we say 'dominated', we mean that *P. rus* is the most common single taxa found on those reefs, not necessarily that it accounts for more than 50% cover. While reports of phase shifts from coral to macroalgae-dominated reefs are more common, studies from Micronesia and the Pacific region have observed phase shifts to weedy coral species [14], sponges [15, 16] and corallimorphs [14, 17].

A report on the state of the sewage system in Tarawa suggests that the benthic community's shift to *P. rus*-dominance could have been begun as early as the mid-1980s, when installation of the sewage pipes damaged corals along the reef flats and crests (60). Unlike many other corals, *P. rus* is tolerant of nutrient loading (63,83), turbidity (84), and heat stress [60–62]. The percent cover of *P. rus* remained relatively stable between 2012 and 2018 (representing on average between 28.31 ± 13.97% in 2012 to 26.51 ± 19.03% in 2018 across sites in Tarawa), although there was a decline from 2012 to 2014 due in part to previously mentioned sampling issues. This decline in *P. rus* between 2012 and 2014 could have also been in part the result of CoTs predation, although we did not find much evidence to support this. We only observed one example of CoTs feeding in Tarawa (and recorded only six CoTs scars, all from a single site), although this may have been the result of our sampling method, because scars are easier to identify on massive *Porites* than *P. rus*.

Research discussing phase shifts on coral reefs often consider recovery to the 'original' state, or phase shift reversals, as a goal for conservation efforts, while noting that the nature of phase shifts makes this difficult [63, 64]. Reversing a phase shift requires addressing the underlying drivers of change, which may ultimately create conditions that facilitate the natural recovery of reefs [64]. One of the most pressing and long-standing issues for reefs in Tarawa is the effects of sewage on the local water quality. In other places, when sanitation systems were improved, local reefs were able to recover from local degradation, although it took several decades before the effects were realized. For example, in Kāne'ohe Bay, Oahu, Hawai'i, reefs shifted from coral dominance to communities dominated by the macroalgae *Dictosphaeria cavernosa* after sewage was discharged into the bay. Like Tarawa, reefs in Kāne'ohe Bay also experienced other stressors, including dredging and siltation. In 1977, the sewage was diverted out of the bay,

and by 2006, *D. cavernosa* had virtually disappeared. Since then, coral cover has increased, and after 44 years of research, researchers have declared that this a successful example of a phase shift reversal [65–67].

While this example shows that removing sources of nutrient pollution can allow reefs to recover from degradation over long time scales, if the relatively stable benthic communities found in Tarawa have entered an alternative stable state, long-term recovery may be more challenging than simply removing the source. However, the definition of an alternative stable state is still being debated. Dungeon et al (2010) define them as occurring when ecosystems exhibit hysteresis (i.e. more than one state can exist under the same environmental conditions at different times). Per this definition, ecosystems with multiple stable states cannot be restored by simply reversing the stressor causing the system to shift into an alternative stable state [68, 69]. Others define alternative stable states as community changes resulting from trends of environmental change [40]; by this definition, there is no difference between an alternative stable state and a phase shift. There is also debate over whether alternative stable states exist at all in nature [70].

Regardless of how alternative stable states are defined (and whether they exist), as Fung et al. [71] point out, tackling multiple human stressors simultaneously can maximize coral reef resilience to phase shifts. Because we have no way of empirically testing whether sites in Tarawa have undergone phase shifts or entered an alternative stable state, we do not speculate further here. Reversing all the threats facing coral reefs in Tarawa will not be possible; coral reef resources are integral for local food and economic security, and while the water quality will likely improve after the sewage system updates, some nutrient pollution is unavoidable. Because of the 'wicked' nature [72] of human-related coral reef degradation in Tarawa, direct human intervention (via coral transplantation projects, for example) will likely be required to change the state of local reefs regardless of whether they entered an alternative stable state and/ or have undergone a phase shift.

Understanding the trade-offs associated with the shift to *P. rus*-dominant reefs will therefore be integral for local decision makers who are tasked with conserving reefs and the ecosystem services that they provide in S. Tarawa. For example, a reef dominated by a single species like *P. rus* may be less able to protect shorelines from wave activity than reefs that are more diverse and are home to a wider range of coral morphologies. That said, recent work from these sites found that there was not a significant difference in the rugosity (or structural complexity) of reefs between Abaiang and Tarawa [73], and complexity is key to reefs' ability to protect shorelines [74]. This difference in complexity across atolls is likely because of the low coral cover in Abaiang [73], and if reefs in Abaiang are able to regain coral cover and larger colonies, we may see that reefs in Abaiang become more structurally complex than those in Tarawa in the future.

## Abaiang

In Abaiang, where the population is smaller compared to the urban communities found along the southern rim of Tarawa, *P. rus* is rare or absent at all sites (Table 1). Sites in Abaiang do not experience the same influx of nutrients and sediments that are found in Tarawa, nor do they experience high fishing pressure. Our results supported our first hypothesis, that post-disturbance communities shifted to dominance by disturbance-resistant benthic taxa. After the 2004–2005 bleaching event, coral reef communities in Abaiang were dominated by relatively thermally tolerant coral taxon such as massive *Porites*. They may have remained in this state after bleaching had they not experienced further disturbance, but the 2013–2014 CoTs outbreak occurred, along with 2009–2010 bleaching event, likely drove the shift in benthic

communities toward a turf-dominated state. As a result, coral cover dropped to the lowest measured since the 2004–2005 bleaching event 11.47 ± 6.86%, with half the sites below the 10% cover threshold proposed for reefs to grow fast enough to keep up with rising sea levels [75]. Although coral cover increased to 18.26 ± 7.45 percent in 2018, one of the sites (ABG002) remained below the 10% threshold.

As hypothesized, the CoTs outbreak disproportionately affected massive *Porites* in Abaiang, one of the 'winners' after the 2009–2010 bleaching event. The results show that massive *Porites* has declined in terms of the percent of the benthos it covers, and the size-structure of the community has shifted towards smaller size classes, which may indicate fragmentation due to partial mortality from the CoTs outbreak. This explanation is consistent with other studies, which found that CoTs outbreaks commonly cause partial mortality of coral colonies [76]. The fragmentation of massive *Porites* could have long-term implications for coral communities in Abaiang. For example, smaller corals of reproductive age release less gametes, which will likely slow recovery of massive *Porites* populations [77]. Massive *Porites* are also slow-growing, stress-tolerant corals, and recovery from disturbance therefore takes longer than it might for faster-growing genera like *Acropora* (although fast-growing, competitive corals also tend to be less resilient to environmental perturbations; this could explain why *Acropora* were rare in Abaiang) [13].

Previous analyses found that the percent cover of *Pocillopora* declined significantly between 2004 and 2012 [19], which may have made massive *Porites* more vulnerable to CoTs predation in 2014. *Pocillopora* is often one of the 'losers' of coral bleaching events, while massive *Porites* are more tolerant of heat stress and are more likely to be 'winners'; this is consistent with what Donner and Carilli (2019) observed when investigating the impacts of the 2004–2005 and 2009–2010 bleaching events. *Pocillopora*, a branching coral, also may indirectly protect massive *Porites* from predation by CoTs. Both *Acropora* and *Pocillopora* are preferred food of CoTs, and thus CoTs will consume those genera over massive *Porites* when there is plenty of prey available [37, 38]. CoTs will actively avoid feeding on massive *Porites* unless its preferred foods are scarce [78].

We found some evidence that reefs in Abaiang could be beginning to recover from coral loss after bleaching and the CoTs outbreak. Although the cover of the many common taxa remained low (as a percent) in all years, the mean percent cover of *Pocillopora* in Abaiang increased by over 60% between 2014 and 2018 (from 1.09 ± 0.88% in 2014 to 2.90 ± 1.65% in 2018), while the mean cover of Favids increased by 88% (from 0.28 ± 0.27% in 2014 to 2.35 ± 1.25% in 2018), and *Montipora* increased by 85% (from 0.09 ± 0.11% in 2014 to 0.58 ± 0.27% in 2018). Overall, the mean live coral cover in Abaiang in 2018 increased by almost 40% between 2016 and 2018 (to 18.26 ± 7.45% in 2018, from 11.47 ± 6.86% in 2016). We had originally planned to repeat our benthic surveys in 2020, but our plans were delayed indefinitely because of COVID; future surveys will hopefully help to further untangle the current and future trajectories of reefs in Abaiang.

We have suggested three potential recovery scenarios for reefs in Abaiang, which are currently dominated by turf algae, in the absence of future acute stressors. In one scenario, the reefs follow a trajectory similar to that experienced by a reef in Moorea, where researchers were able to observe the entire trajectory of a CoTs outbreak [79]. Like what we observed from Abaiang, turf algae were the first taxa to colonize the empty spaces left after CoTs had decimated coral populations, but coral dominance returned about a decade post-disturbance, and the communities went from coral-dominant to turf algae-dominant, and then back to coral-dominant. Under this scenario, Abaiang's communities recover to become dominated by thermally-tolerant species such as massive *Porites* and *Heliopora*, along with some fast-growing species, such as *Pocillopora* but likely not *Acropora*. Because *Pocillopora* is a brooding species,

its recovery is not density-dependent like most *Acropora* species, which are vulnerable to Allele effects [77, 80, 81]. Corals of the *Acropora* genera were rare in Abaiang prior to the study period [19], and bleaching and CoTs both disproportionately negatively affect *Acropora* [39].

Alternatively, it is possible that reefs in Abaiang will experience a shift towards *Halimeda*-dominance in the future (with *Halimeda* cover exceeding that of turf algae), if the local reef fishery is enough to limit herbivory on reefs; *Halimeda* is vulnerable to predation from common herbivorous reef fish, in particular *Acanthuridae* and *Scaridae* [82]. The data suggest this is unlikely, given that *Halimeda* has declined significantly over the current study period (Table 2). Indeed, the case study from Moorea suggests that a phase shift to macroalgae dominance would require an additional disturbance post-CoTs, such as a reduction in grazing due to fishing pressure or poor water quality [79]. We have no evidence that either of these two conditions currently exist in Abaiang.

The third potential scenario we propose for reefs in Abaiang is that they remain in their current state. If future acute disturbances occur in Abaiang, they may prevent coral communities from recovering. However, we also find it unlikely that benthic communities in Abaiang will remain dominated by turf-algae given that the percent cover and size frequency of corals have both changed after disturbance in Abaiang. A previous study found that massive *Porites* in the Gilberts that survived bleaching in 2004–2005 were less susceptible to bleaching in 2009–2010 [18]. This suggests that the corals remaining in Abaiang that survived both bleaching and CoTs are less likely to bleach during future heat stress events.

## CoTs outbreak

Previous research suggested that low-latitude coral reefs might be less susceptible to CoTs outbreaks because CoTs are not tolerant of SSTs that are higher than 30˚C [66, 83]. Survivorship, particularly of larvae and juveniles, declines above 30˚C, and temperatures above 29˚C can negatively impact embryonic and larvae development [37]. Extended La Niña conditions lowered SSTs in the Gilberts from 2010–2013, during which SST at our sites averaged 28.62˚C, just below the 29˚C temperature threshold for negative impacts on CoTs larvae development (for comparison, the mean SST for 2010–2018 was 29.08˚C). These slightly cooler-than-average conditions could have facilitated the survival of CoTs larvae, contributing to the outbreak in 2013–2014. However, CoTs may be able to adapt to increasing SSTs [84], in which case Tarawa and Abaiang may be vulnerable to more CoTs outbreaks in the future. We suggest that future studies investigate the potential links between ENSO events and CoTs outbreaks in the central Pacific.

The CoTs affected sites in Abaiang disproportionately, and we are unable to account for the different severities of the outbreak across atolls. While larvae may have reached Tarawa at the same time as Abaiang, they may not have been as successful at settling and/or reaching maturity in Tarawa, but it is unlikely that this differential survivorship would be due to excess nutrients in Tarawa; on the Great Barrier Reef, researchers have found that CoTs outbreaks are positively correlated with high levels of nutrients [85, 86]. CoTs will actively avoid feeding on *P. rus* in favor of other corals, but they will feed on less-preferred prey when their preferred food items are scarce [7, 67, 87].

Because we were not able to conduct reef surveys at beginning of the outbreak, these data represent a limited snapshot of the event at one point in time. Reports to the MFMRD from the outer atolls suggest the outbreak was widespread, stretching from Butaritari Atoll (3˚N) to at least Abemama Atoll (~0˚), where a related team observed a CoTs outbreak during fieldwork in October–November, 2013 [35]. Synchronous outbreaks of CoTs over wide distances has occurred before [88]; still, even if we assume that all sites in Abaiang were affected by

CoTs, we are unable to say whether all sites experienced the same levels of predation and/or outbreak duration. CoTs larvae may have been more likely to settle and survive at some sites than others. Their survival and impact at specific reefs may have varied by local conditions, including oceanography, water quality, and the amount and type of coral prey available and quality. In addition, the CoTs scars we counted in Abaiang are likely underestimated because we only counted scars that were recent (as indicated by the visibility of the coral skeleton); any scars that had already been colonized by turf algae or other taxa were excluded because we could not say positively how old they were or that they were not caused by other factors. Many of the dynamics of CoTs larvae settlement and their survival post-settlement are largely unresolved [37].

### Disturbance, reef health, and resilience

Our sites at both atolls exhibit characteristics that are the opposite of what we would expect if we had relied on the most common metrics of disturbance on coral reefs, for example, the percent of all macroalgae or the percent of all live coral cover. For example, high macroalgae cover is often used by coral reef researchers to quantify degradation or to distinguish between 'healthy' and 'unhealthy' reefs [12]. However, macroalgae was most common on reefs in Abaiang (and had very low cover in Tarawa). The percent cover of the macroalgae genera *Halimeda* has declined over time in Abaiang, while turf algae has increased. Sites in Abaiang had lower coral cover than sites in Tarawa; if we had chosen to consider the percent cover of all live coral a metric for reef health, we may have mistakenly concluded that sites in Tarawa are healthier and less degraded than those in Abaiang. That said, while reefs in Abaiang are arguably less degraded than reefs in Tarawa (that is, less impacted by local human disturbance), we also would not necessarily consider them 'healthy,' given that the community composition in Abaiang at the end of our study period is the result of repeated acute disturbances from which they have yet to recover. Our past work in the Marshall Islands similarly found that using broad categories of taxa like macroalgae and live coral to classify the health of coral reefs could be misleading and lead to incorrect conclusions [12]. Here, we echo that call to use more concise metrics and language when discussing the state of coral reefs facing multiple stressors at different scales.

This work may support the hypothesis proposed by other reef scientists that more degraded reefs may be more resistant to the impacts of climate change [89, 90], which is contrary to the argument that controlling local stressors could improve resistance to and recovery from temperature stress and other acute disturbances [89]. For example, sites in Tarawa were more impacted by local stressors than sites in Abaiang, but the coral cover has remained higher in Tarawa. The percent cover of *P. rus* did not change significantly over time in Tarawa, indicating that *P. rus* was largely unaffected by both heat stress and the CoTs event. There is evidence that *P. rus* is indeed insensitive to heat stress [60–62], and is also not significantly affected by short-term exposure to high pCO$_2$ [91]. Combined with the evidence that *P. rus* is also resilient to localized human impacts such as nutrient loading and turbidity, we find it likely that *P. rus* will be able to outcompete other species that are more sensitive to multiple stressors in the future, particularly if local impacts continue unabated (although the sewage upgrades will likely reduce nutrient loading). Unlike other corals that are sensitive to high pCO$_2$, increased acidification does not appear to negatively impact calcification rates of *P. rus*, and researchers do not expect that the calcification of this species will be affected strongly by the projected increase in pCO$_2$ that is expected to occur by the end of the century [91].

However, for a coral reef to be considered resilient, it must be both resistant to disturbance and be able to recover to the original community structure post-disturbance, without an

associated loss in function and services [90]. We have provided further evidence that the *P. rus*-dominated reefs in Tarawa are more resistant to heat stress than those in Abaiang, which is in agreement with previous studies [19]. Sites in Abaiang may become more resistant to heat stress in the future, depending on their recovery trajectory (if, for example, coral cover increases and is composed of more thermally tolerant genera). Also, because sites in Tarawa were dominated by a single coral species (*P. rus*), they are potentially vulnerable to future 'ecological surprises' [92]; any disturbance that has a disproportionate impact on *P. rus* could have a severe impact on the ecosystem as a whole. There was also likely a loss in function and services associated with the phase shift to *P. rus* at sites in Tarawa, which indicates that while these reefs are resistant to heat stress, they may not be resilient. Surveys of fish and invertebrate assemblages, as well as and reef structural complexity, would provide useful information about how the phase shift to *P. rus* is influencing the ecosystem services that are valuable to people residing in Tarawa.

Coral reefs in the Gilbert Islands have experienced years with prolonged heat stress more frequently than 99% of the world's coral reefs [19], but this may change in the future; other reefs will likely experience more heat stress going forward, given that climate change-driven global coral bleaching events have increased in frequency and are expected to continue increasing in the due to climate change [93]. Reefs in the Gilbert Islands could therefore provide a rare glimpse into what reefs may look like in the future while also accounting for a gradient of local human impacts. We hope this work, coupled with further investigations into the potential trade-offs associated with locally degraded but climate-resistant reefs such as found in Tarawa versus reefs with more long-term ability to recover but with less resistance such as those in Abaiang, will provide novel information that is important for the future management of coral reef resources. Specifically, these findings may help to predict the ways that climate change will affect the millions of people around the world who depend on coral reefs, so that they may prepare for the future.

## Supporting information

**S1 Table. Environmental variables, coordinates, and years data were collected for 19 study sites visited between 2012 and 2018.**
(DOCX)

**S2 Table. Coral order, family, and species observed in Tarawa and Abaiang atolls (as reported by Lovell et al, 2000 [30]).**
(DOCX)

**S3 Table. Results of linear mixed effects models for each key benthic category, including additional LMMs for subsets of the data.** Statistically significant results at α = 0.05 are in bold, while those that are significant at α = 0.10 are underlined.
(DOCX)

**S4 Table. Tukey results for linear mixed effects models, percent ~ year + (1|site).** All p-values have been adjusted for multiple comparisons. Results significant at sigma = 0.05 are in bold; those significant at sigma = 0.10 are underlined.
(DOCX)

**S5 Table. Results of size frequency statistical analyses.** Includes Welch's ANOVA of size-frequency statistics between years, and KS test results comparing size frequency distributions across years within each atoll.
(DOCX)

## Acknowledgments

This work would not have been possible without Tooreka Teemari, Karibanang Tamuera, and all our colleagues at the Kiribati Ministry of Fisheries & Marine Resource Development and the Ministry of Environment, Lands, and Agricultural Development. We also thank the Kiribati Meteorological Station for providing local wind vector data. We are grateful to Heather Summers for her help and comradery during fieldwork, Pedro Gonzalez for collecting and extracting the satellite data, Eric Leinberger, who produced the map of our research sites, and our undergraduate student assistants Katrina Bernaus, Alex Tso and Steuart Tannason for their work processing quadrat photos and conducting the NDVI calculations. Finally, we thank editor Dr. James Guest and the anonymous reviewers whose comments and suggestions made this work stronger. We would also like to acknowledge that all statistical analyses were conducted on the unceded, traditional, and ancestral territories of the xʷməθkʷəýəm (Musqueam) people, in what is now called Vancouver, British Columbia, Canada.

## Author Contributions

**Conceptualization:** Sara E. Cannon, Simon D. Donner.

**Data curation:** Sara E. Cannon, Simon D. Donner.

**Formal analysis:** Sara E. Cannon.

**Funding acquisition:** Simon D. Donner.

**Investigation:** Sara E. Cannon, Erietera Aram, Toaea Beiateuea, Aranteiti Kiareti, Max Peter.

**Methodology:** Sara E. Cannon, Simon D. Donner.

**Project administration:** Sara E. Cannon, Simon D. Donner.

**Resources:** Simon D. Donner.

**Supervision:** Simon D. Donner.

**Validation:** Sara E. Cannon.

**Visualization:** Sara E. Cannon.

**Writing – original draft:** Sara E. Cannon.

**Writing – review & editing:** Sara E. Cannon, Erietera Aram, Toaea Beiateuea, Aranteiti Kiareti, Max Peter, Simon D. Donner.

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
