## [Decision Letter · Decision Letter 0]

9 Mar 2021

PONE-D-21-04308

Coral reefs in the Gilbert Islands of Kiribati: Resistance, resilience, and recovery after more than a decade of multiple stressors

PLOS ONE

Dear Dr. Cannon,

Thank you for submitting your manuscript to PLOS ONE. After careful consideration, we feel that it has merit but does not fully meet PLOS ONE’s publication criteria as it currently stands. Therefore, we invite you to submit a revised version of the manuscript that addresses the points raised during the review process.

The manuscript was sent to two reviewers and they both gave detailed and fairly similar comments. They both agreed that there's some really great data here that are worthy of publication, but also agreed that the paper needs to be shortened and drastically streamlined. Reviewer 2 gives some really useful advice on how to do this. I urge you to take on board these comments and address each in turn before submittinfg your revision.

Please submit your revised manuscript within 90 days. If you will need more time than this to complete your revisions, please reply to this message or contact the journal office at plosone@plos.org. Please include the following items when submitting your revised manuscript:

We look forward to receiving your revised manuscript.

Kind regards,

James R. Guest, Ph.D.

Academic Editor

PLOS ONE

Additional Editor Comments:

I note that you are using the term Favid as a taxon name. I suggest looking at the most up to date taxonmy of corals. My understanding is that Faviidae has changed to Merulinidae (see papers by Danwei Huang and others for the most updated revisions of the Merulinidae, or Bigmessidae as they call it!)..

Journal Requirements:

4. We note that Figure 1 in your submission contain map images which may be copyrighted. All PLOS content is published under the Creative Commons Attribution License (CC BY 4.0), which means that the manuscript, images, and Supporting Information files will be freely available online, and any third party is permitted to access, download, copy, distribute, and use these materials in any way, even commercially, with proper attribution. For these reasons, we cannot publish previously copyrighted maps or satellite images created using proprietary data, such as Google software (Google Maps, Street View, and Earth). For more information, see our copyright guidelines: http://journals.plos.org/plosone/s/licenses-and-copyright.

4.1.    You may seek permission from the original copyright holder of Figure 1 to publish the content specifically under the CC BY 4.0 license. 

4.2.    If you are unable to obtain permission from the original copyright holder to publish these figures under the CC BY 4.0 license or if the copyright holder’s requirements are incompatible with the CC BY 4.0 license, please either i) remove the figure or ii) supply a replacement figure that complies with the CC BY 4.0 license. Please check copyright information on all replacement figures and update the figure caption with source information. If applicable, please specify in the figure caption text when a figure is similar but not identical to the original image and is therefore for illustrative purposes only.

Reviewers' comments:

Reviewer's Responses to Questions

**Comments to the Author**

1. Is the manuscript technically sound, and do the data support the conclusions?

Reviewer #1: Yes

Reviewer #2: Yes

2. Has the statistical analysis been performed appropriately and rigorously? 

Reviewer #1: Yes

Reviewer #2: Yes

3. Have the authors made all data underlying the findings in their manuscript fully available?

Reviewer #1: Yes

Reviewer #2: Yes

4. Is the manuscript presented in an intelligible fashion and written in standard English?

Reviewer #1: Yes

Reviewer #2: Yes

5. Review Comments to the Author

Reviewer #1: This study addresses coral reef community dynamics in the Gilbert Islands, which represents a remote corner of the world with fascinating lessons to be learned about coral reef ecology that will benefit understanding of coral reefs throughout the world. Cannon and colleagues have accomplished an impressive task of surveying these reefs and overcoming challenging logistics at the end of a very long supply chain from Canada.

Like many coral reef studies, this analysis seeks to report on the recovery of reefs following major disturbances, and it uses multiple surveys conducted from 2012 to 2018 to describe changes in cover and changes in size frequency structure of common coral genera. The effort to go beyond reporting coral cover is laudable! The content of this paper is interesting and worthy of publication, but the length of the presentation, the complexity of data, and the weak focus greatly detracts from comprehension. This type of material does not require a 10 page introduction and 66 references to set the scene, and the discussion needs to address broader issues beyond the confines of the study locality. Likewise, the presentation of the results in 11 tables and 7 figures strongly argues for a great reduction in scope, a focus on testable hypothesis, and a more judicious selection of salient data to convey the key outcomes. This process might lead to the logical conclusion that the study should be split into multiple papers.

Intellectually, the context of the paper is provided by phase change theory and how this has been applied to coral reefs. This material is fascinating and appropriate, but it is only part (perhaps only half) of the critical context: what about alternative stable state theory? While this is a topic that gains much attention and debate, and researchers tend to be polarized in their opinion, discussion of how coral reefs have changed from coral to macroalgae really is not complete without some treatment of alternative stable states. Critically, if the changes affecting the reef represent an alternative stable state, then simply reversing the disturbances that caused the change will be insufficient to restore the initial state. Obviously this has important implications for conservation and recommendations for management.

Reviewer #2: This manuscript evaluated changes in benthic community structure on Tarawa and Abaiang Atolls in relation to warming events and a Crown of Thorns outbreak. The authors compare communities at multiple levels of human disturbance (which they calculate two ways). They also investigate some oceanographic drivers that may be influencing benthic community structure.

Overall, I think that the information presented in this manuscript is important for understanding trajectories of ecosystem change in central Pacific reefs. I do think that the manuscript should be shortened and tightened up considerably. Both the Introduction and the Discussion were a bit meandering, and therefore somewhat hard to follow. I think that this should be published, but that improvements should be made to clarify the structure and story throughout. I have included some specific suggestions below.

ABSTRACT

Please include some percent cover values in the abstract. Much of the paper is focused on comparing percent cover of different taxa at different locations, but these results should be explicitly mentioned in the abstract. The information in lines 694-698 would be a good candidate here.

INTRODUCTION

- You mention that the first bleaching event recorded at Tarawa and Abaiang occurred in 2004-2005, is there any local knowledge that would suggest if bleaching events have been noticed before?

- I actually liked the depth that you went into regarding the history and changes that have occurred in the Gilberts. Usually I'd recommend cutting down the intro quite a bit, but instead I would like to recommend that you try to streamline it a bit more. This will shorten it, and can also help with the flow of the manuscript. Focus on the topic sentences, and then tighten each paragraph up a bit.

METHODS

- There are *a lot* of tables in this manuscript. Which is great (!) because you have so much data to share. However, I think that some of them should be moved to the Supplementary Material, so you only retain those which are most important to your main findings.

- Table 1 - please include a brief description of each of the included metrics

- Would it be possible to add some text regarding which species of coral are expected on these atolls. Specifically, you list percent cover at the genus level, but it would be helpful to know what species might be present. I assume that this wasn't collected during sampling, but with all your research there, perhaps you have a species list you could include as a supplement? This would be particularly useful for "Favids" as this is potentially a very broad taxon.

- Note that NOAA calculates MMM based on 1985-1993 but excludes the years 1991 and 1992, they don't calculate based on 1985-1994.

- I was curious about how currents might influence your analysis described in lines 442-446. If currents are strong or directional, it might matter less how far away you are from a sewer outfall and more whether you're downstream of the outfall.

RESULTS

- Lines 516-532 - I think that it would be a good idea to run these analyses with the two atolls separately along with the grouped analysis that you've already done. I think the version in the manuscript is informative, but I am curious about how presence/absence of certain taxa on each atoll affects the interpretation of these results. Breaking it up by atoll would also allow you to look at finer scale change over time. If you did do this, and I somehow missed it, it would be helpful to improve clarity about which analyses were done, as this was a bit hard to follow.

- Why did you decide to use annual mean SST versus maximum monthly mean or similar?

- The results of the PERMANOVA as reported a little unconventionally. Specifically, in line 606 "The factor contributing most to the variation in the percent cover was 'year and mean NDVI'". I believe this should be reported as "the interaction between year and mean NDVI". That is, percent cover changes differently across years based on mean NDVI. This should be more clearly stated.

- I also think that the PERMANOVA should be run again, and separately for each island. I am not convinced that there isn't bias based on which locations were sampled during each field season, and would be convinced if you found similar results with individual models. I noticed that you ran the SIMPER for each atoll specifically, which I think was a good idea.

DISCUSSION

- The discussion is too long, and should be streamlined considerably. One thing that would be helpful would be careful thought toward the overall organization, and trimming unnecessary information that doesn't speak directly to your results and conclusions. For example, the paragraph from lines 755-769 is mostly unnecessary.

- Lines 1037-1038 - "we therefore do not find it likely that the presence of Halimeda in Abaiang was triggered by any acute disturbances that allowed the macroalgae to outcompete corals." I do not think this is well supported by the results, as you don't know the history before the 1990s, and whether the reefs did (or did not) rapidly change from coral dominance to Halimeda dominance.

- When talking about the percent cover metrics in the context of calling a reef "healthy" or "degraded" I think the point (that percent coral doesn't tell you everything) is good (e.g. in lines 1038-1045), however, in the following paragraph (lines 1047-1061), you label those reefs dominated by P. rus to be "resilient". I think it would be worthwhile to consider, and explicitly define, what makes a reef resilient. If a reef with relatively high coral cover isn't "healthy" (e.g., sites at Tarawa), is it actually resilient? Sure, it is resilient as it seems like this may be a stable state that the ecosystem tends towards under these conditions, but under that definition, an algal dominated reef is also resilient, just toward a different state. Usually in this context resilient means something more - that reefs will maintain their structure and function - which is likely not true at Tarawa. I think this is an important point that needs to be discussed in the manuscript.

FIGURES

Figure 3 - Is it possible to include earlier dates for historical warming at this location?

Figure 4 - Since you mention the difference between macroalgae and turf, would it be possible to include turf on both of these panels, in addition to live coral and macroalgae?

Figure 5 - Pocillopora is spelled incorrectly

EDITS

Line 20 - change "in" to "on"

Line 26 - change "The" to "These"

Line 27 - remove "the"

Lines 65-66 - I'm not convinced about saying "most commonly" here, and I'd suggest removing it

Line 122 - rephrase "more recent work questions that conclusion" to "this conclusion is still under debate"

Line 202 - add in "Hawaii" after Kane`ohe Bay

Line 217 - replace "continue" with "continued"

Line 219 - remove "researchers found that"

Line 220 - replace "had been" with "was"

Line 224 - replace "the" with "these"

Line 260 - does "ocean side" mean "reef slope" please be a little more specific

Line 375 - remove "the" from "For the percent cover"

Line 470 - Table 2 - what is "other morphology/species" for Porites?

6. PLOS authors have the option to publish the peer review history of their article (what does this mean?). If published, this will include your full peer review and any attached files.

Reviewer #1: No

Reviewer #2: No

---

## [Author Response · Author response to Decision Letter 0]

23 Apr 2021

21 April, 2021

Re: PONE-D-21-04308, “Coral reefs in the Gilbert Islands of Kiribati: Resistance, resilience, and recovery after more than a decade of multiple stressors”

Dear Dr. Guest,

Thank you for the detailed and helpful feedback and the two anonymous reviews you have shared with us for our recent manuscript submission to PLOS ONE. On behalf of my co-authors, I am submitting a revised manuscript that responds to the very helpful suggestions and concerns raised by the reviewers.

In response to the reviewer’s comments, we have significantly shortened the manuscript (by roughly 6,000 words), stated our tested hypotheses in the introduction, added a more complete discussion of alternative stable states, removed some of the superfluous analyses, and made a number of other small changes (as detailed further below). The manuscript is now a concise summary of this research and how it fits into the relevant existing literature.

The reviewers disagreed about the inclusion of the history of Kiribati in the introduction. We have decided to shorten the history but to keep it in the manuscript because we deemed it important that we discuss the colonial drivers of current stressors on these reefs. While we needed to discuss the stressors existing locally, we could have done this without the historical context. However, this context is integral to avoid implying that these stressors were caused by local mismanagement, which could inadvertently bolster the perception that local managers are not equipped to manage marine resources themselves. Unfortunately, our personal experience tells us that this perception is not uncommon among researchers working in this part of the world. Instead, the current states of these ecosystems are the result of processes that were outside of the control of the people who are responsible for addressing them today. That said, we agree with the feedback that you and both reviewers provided that the introduction was long and difficult to follow. We have made changes to streamline the introduction in response to the suggestions and are grateful that this feedback has helped us to make the manuscript more concise and intelligible.

I am including my responses to each of the comments below (the original comment is in regular text and my response is in italics). I have highlighted where changes were made in the manuscript to address the comments and have also uploaded two versions of the revised manuscript (one that tracks the changes, and one that does not), per your instruction.

Thank you again for your response and suggestions. We are grateful for the opportunity to respond and look forward to hearing back from you.

Sincerely,

Sara E. Cannon, M.Sc., Ph.D. Candidate

University of British Columbia

Response to Reviews

Additional Editor Comments:

I note that you are using the term Favid as a taxon name. I suggest looking at the most up to date taxonomy of corals. My understanding is that Faviidae has changed to Merulinidae (see papers by Danwei Huang and others for the most updated revisions of the Merulinidae, or Bigmessidae as they call it!).

Thank you for highlighting these changes in coral taxonomy. While we used photos to calculate the percent cover and could therefore re-identify anything categorized as a ‘Favid’ to reflect the most current taxonomy, we identified the corals for the size frequency analyses in situ. Some corals that were difficult to identify to the genus level were identified at the family level during data collection; also, within the Favid family, some of the species have since been split into different genera and families (we did not identify corals to the species level, with the exception of Porites rus). Unfortunately, we are therefore unable to correct those data for the updated taxonomy. We have updated the text of the manuscript in the methods section to note the changes in taxonomy, and to clarify that we are using the taxonomy from Veron (2000) Corals of the World which do not reflect these changes, along with our reasoning for doing so [1], as we have seen done in other recent papers [for example, see 2].

Thank you for this reminder. We will make sure we are careful to follow the file naming convention and the style requirements in the future.

We have added this information to the manuscript. This research was made possible by an ongoing relationship between S. Donner and the Coastal Fisheries Division of the Ministry of Fisheries and Marine Resource Development, which emerged out of the Kiribati Adaptation Project in 2007. All surveys were conducted with our MFMRD co-authors and are coupled with their work activities. Prior to 2018, the surveys were treated as a part of MFMRD activities and thus we did not require a permit. For the 2018 surveys, Cannon and Donner requested and secured permission from the Ministry of Environment, Lands, and Agricultural Development (MELAD) to meet new regulations (they do not issue official permits or permit numbers).

Thank you. We would prefer to release the data in the event that the manuscript is accepted for publication and are aware that PLOS ONE will hold the manuscript until this is completed. We have already uploaded the data to Zenodo, although they are currently not shared publicly. We will make them public should the manuscript be accepted. 

4. We note that Figure 1 in your submission contain map images which may be copyrighted. All PLOS content is published under the Creative Commons Attribution License (CC BY 4.0), which means that the manuscript, images, and Supporting Information files will be freely available online, and any third party is permitted to access, download, copy, distribute, and use these materials in any way, even commercially, with proper attribution. For these reasons, we cannot publish previously copyrighted maps or satellite images created using proprietary data, such as Google software (Google Maps, Street View, and Earth). For more information, see our copyright guidelines: http://journals.plos.org/plosone/s/licenses-and-copyright.

We require you to either (1) present written permission from the copyright holder to publish these figures specifically under the CC BY 4.0 license, or (2) remove the figures from your submission.

Thank you for this inquiry. Fig 1 does not contain any imagery that is protected by copyright, although we did overlook two citations for the data, which should have been included in the original manuscript. We apologize for this oversight and have added the required information to the manuscript, in both the figure caption and the manuscript.

Reviewers' comments:

Reviewer's Responses to Questions

Comments to the Author

Reviewer #1: This study addresses coral reef community dynamics in the Gilbert Islands, which represents a remote corner of the world with fascinating lessons to be learned about coral reef ecology that will benefit understanding of coral reefs throughout the world. Cannon and colleagues have accomplished an impressive task of surveying these reefs and overcoming challenging logistics at the end of a very long supply chain from Canada.

Like many coral reef studies, this analysis seeks to report on the recovery of reefs following major disturbances, and it uses multiple surveys conducted from 2012 to 2018 to describe changes in cover and changes in size frequency structure of common coral genera. The effort to go beyond reporting coral cover is laudable! The content of this paper is interesting and worthy of publication, but the length of the presentation, the complexity of data, and the weak focus greatly detracts from comprehension. This type of material does not require a 10 page introduction and 66 references to set the scene, and the discussion needs to address broader issues beyond the confines of the study locality. 

Thank you for this helpful feedback and for your suggestions. We are grateful for the time you have taken to share your expertise in order to strengthen this manuscript. 

We have made several changes to the introduction in order to streamline the manuscript. The original introduction was 2,625 words with 66 references, and it is now 1,669 words with 40 references. We have also incorporated your suggestion to focus on testable hypotheses (Lines 163-173), and we agree that doing so has helped to improve the focus of the paper (specifically, it helped us to highlight which analyses could be removed without detracting from the overall findings).

Specifically, we hypothesized that 1) post-bleaching communities shifted towards dominance by disturbance-resistant coral taxa and macroalgae over the study period, 2) that the shift to P. rus in S. Tarawa is persistent and represents a phase shift, 3) that the taxa-level response to the CoTs outbreak differs from that of bleaching, and finally, 4) that the trajectories of post bleaching communities differed based on human disturbance.

The two reviewers disagreed on the inclusion of the historical context. As mentioned in the cover letter, we opted to retain the majority of this material in order to counter misconceptions about local capacity to manage coral reef resources. However, we condensed this material significantly and reordered the introduction significantly in response to your comments.

Likewise, the presentation of the results in 11 tables and 7 figures strongly argues for a great reduction in scope, a focus on testable hypothesis, and a more judicious selection of salient data to convey the key outcomes. This process might lead to the logical conclusion that the study should be split into multiple papers.

We are grateful for this feedback. We have taken your suggestion to reduce the scope of the paper by removing several analyses that upon reflection, we felt detracted from the key themes of the paper. We also re-organized the paper to focus on four testable hypotheses, per your suggestion (which further supported our decision to reduce the number of analyses). 

Specifically, we removed the analysis of P. rus cover and distance to the sewage pipes in S. Tarawa and the linear mixed effects model to test for a relationship between the percent cover of Halimeda and sea surface temperature. We also removed the Principal Component Analysis because we felt that it did not add any information that was not already incorporated in the SIMPER Analysis and the PERMANOVA. We also reduced the number of tables to seven (moving some to the supplementary materials, for transparency), and the number of figures to five.

Intellectually, the context of the paper is provided by phase change theory and how this has been applied to coral reefs. This material is fascinating and appropriate, but it is only part (perhaps only half) of the critical context: what about alternative stable state theory? While this is a topic that gains much attention and debate, and researchers tend to be polarized in their opinion, discussion of how coral reefs have changed from coral to macroalgae really is not complete without some treatment of alternative stable states. Critically, if the changes affecting the reef represent an alternative stable state, then simply reversing the disturbances that caused the change will be insufficient to restore the initial state. Obviously this has important implications for conservation and recommendations for management.

Thank you for this perspective. We originally decided not to address alternative stable states it in the discussion because we are unable to test whether an alternative stable state exists in Tarawa. We had also hoped to avoid polarizing readers due to the controversy in the literature. However, after receiving your feedback, we have given this further thought and agree that the discussion is incomplete without some mention of alternative stable states. We have updated the manuscript accordingly, per your suggestion.

Specifically, we added two paragraphs to the discussion, beginning at line 688:

“While this example shows that removing sources of nutrient pollution can allow reefs to recover from degradation over long time scales, if the relatively stable benthic communities found in Tarawa have entered an alternative stable state (ASS), long-term recovery may be more challenging than simply removing the source. However, the definition of ASS is still being debated. Dungeon et al (2010) define them as occurring when ecosystems exhibit hysteresis (i.e. more than one state can exist under the same environmental conditions at different times). Per this definition, ecosystems with multiple stable states cannot be restored by simply reversing the stressor causing the system to shift into an alternative stable state [68,69]. Others define alternative stable states as community changes resulting from trends of environmental change [40]; by this definition, there is no difference between an ASS and a phase shift. There is also debate over whether ASS exist at all in nature [70]. 

Regardless of how ASS is defined (and whether they exist), as Fung et al. [71] point out, tackling multiple human stressors simultaneously can maximize coral reef resilience to phase shifts. Because we have no way of empirically testing whether sites in Tarawa have undergone phase shifts or entered an ASS, we do not speculate further here. Reversing all of the threats facing coral reefs in Tarawa will not be possible; coral reef resources are integral for local food and economic security, and while the water quality will likely improve after the sewage system updates, some nutrient pollution is unavoidable. Because of the ‘wicked’ nature [72] of human-related coral reef degradation in Tarawa, direct human intervention (via coral transplantation projects, for example) will likely be required to change the state of local reefs regardless of whether they entered an ASS and/or have undergone a phase shift.”

We then follow these paragraphs with a discussion of the potential trade-offs associated with a shift to P. rus-dominated reefs in S. Tarawa, beginning at line 711.

Reviewer #2: This manuscript evaluated changes in benthic community structure on Tarawa and Abaiang Atolls in relation to warming events and a Crown of Thorns outbreak. The authors compare communities at multiple levels of human disturbance (which they calculate two ways). They also investigate some oceanographic drivers that may be influencing benthic community structure.

Overall, I think that the information presented in this manuscript is important for understanding trajectories of ecosystem change in central Pacific reefs. I do think that the manuscript should be shortened and tightened up considerably. Both the Introduction and the Discussion were a bit meandering, and therefore somewhat hard to follow. I think that this should be published, but that improvements should be made to clarify the structure and story throughout. I have included some specific suggestions below.

Thank you for the feedback. As noted in the response to Reviewer #1, we have substantially reduced the length of the manuscript and focused the introduction and the discussion in particular, although we have also substantially edited the methods and results to reflect the reduced number of analyses.

ABSTRACT

Please include some percent cover values in the abstract. Much of the paper is focused on comparing percent cover of different taxa at different locations, but these results should be explicitly mentioned in the abstract. The information in lines 694-698 would be a good candidate here.

Thank you for this suggestion. We have updated the abstract to include more specifics about our findings, including some percent cover values as you recommended. Specifically, we added the following, beginning on line 31:

“In densely populated South Tarawa, we document a phase shift to the weedy and less bleaching-sensitive coral Porites rus, which accounted for 81% of all coral cover by 2018. By contrast, in less populated Abaiang, coral communities remained comparatively more diverse (with higher percentages of Pocillopora and the octocoral Heliopora) after the disturbances, but reefs had lower overall hard coral cover (18%) and were dominated by turf algae (41%).”

INTRODUCTION - You mention that the first bleaching event recorded at Tarawa and Abaiang occurred in 2004-2005, is there any local knowledge that would suggest if bleaching events have been noticed before?

Thank you for this question. We have asked during our many years of work in Tarawa and Abaiang, but no one has reported mass bleaching prior to 2004. It is very possible that bleaching did occur; a previous analysis [3] examined past climate history using satellite data and found that bleaching-level heat stress did occur prior 2004 in accordance with central Pacific El Niño events. The only bleaching report (for 1998) is based on later measurement of radial growth of Porites microatolls (https://repository.si.edu/handle/10088/4931), and the evidence is limited to the shallow, warm Abaiang lagoon (personal communication with the author, the scientist and teacher Charles Flora, who lived in Abaiang at the time).

However, as noted in [3], surveys of the published literature, grey literature, the online database Reefbase, and local experts and found no reports of bleaching prior to 2004. Additionally, a coral coring study [4] found no evidence of partial mortality scars or significant reduction in growth rates of multiple cores in massive Porites spp. in Abaiang or Tarawa, although the majority of the cores went back only to 1980. Therefore, while it is possible bleaching occurred and did not affect the massive Porites sampled, there are no direct observations. We have updated this in the manuscript beginning at line 132. 

- I actually liked the depth that you went into regarding the history and changes that have occurred in the Gilberts. Usually I'd recommend cutting down the intro quite a bit, but instead I would like to recommend that you try to streamline it a bit more. This will shorten it, and can also help with the flow of the manuscript. Focus on the topic sentences, and then tighten each paragraph up a bit.

Thank you for this insight and we are glad you agree with our inclusion of this history, although we agree with suggestion to streamline the introduction. As mentioned in the response to Reviewer #1, we have taken your advice to reorganize and to tighten the paragraphs, and hope that it has made the introduction more comprehensible.

METHODS

- There are *a lot* of tables in this manuscript. Which is great (!) because you have so much data to share. However, I think that some of them should be moved to the Supplementary Material, so you only retain those which are most important to your main findings.

Thank you for this suggestion. We have deleted Table 2 and moved Tables 8 and 9 to the supplementary materials per your feedback. We now have seven tables, instead of the eleven we had in the original manuscript.

- Table 1 - please include a brief description of each of the included metrics

We have updated the table to include brief descriptions of the metrics, per your suggestion.

- Would it be possible to add some text regarding which species of coral are expected on these atolls. Specifically, you list percent cover at the genus level, but it would be helpful to know what species might be present. I assume that this wasn't collected during sampling, but with all your research there, perhaps you have a species list you could include as a supplement? This would be particularly useful for "Favids" as this is potentially a very broad taxon.

Thank you for this suggestion. We have not identified corals in these locations to the species level, and unfortunately are unable to provide a recent list of species. The last inventory of coral species was collected in the late 1990’s and recorded 127 distinct species of hard corals in Abaiang and Tarawa [5]. We have recreated this list to include it as a supplement (with some minor changes in formatting, because the original table included species that were found at other nearby locations such as atolls in the Marshall Islands) and have referenced it in the manuscript.

- Note that NOAA calculates MMM based on 1985-1993 but excludes the years 1991 and 1992, they don't calculate based on 1985-1994.

Here, we calculated the MMM ourselves using data from 1985-1994, and we recognize that this time period is distinct from how NOAA calculates the MMM (which now is based on centering the climatology around 1987 using regression of SST trends). Thanks to this comment, we realized that the way we worded our methods made it sound as if we were employing the same methods to calculate the MMM that NOAA uses. We have rephrased the methods to avoid this confusion. 

- I was curious about how currents might influence your analysis described in lines 442-446. If currents are strong or directional, it might matter less how far away you are from a sewer outfall and more whether you're downstream of the outfall.

This is an important point and we appreciate that you raised it. The prevailing winds are north/northwest, hence blowing towards the southern and eastern parts of the atolls. We do not have data on the currents at depth, but surface observations suggest some westerly water movement on South Tarawa’s outer reef caused by the winds and the geography of atoll. Before the outflows were moved in 2019, they expelled sewage at approximately 7m depth. The sewage outflows roughly face south, which means that the sewage would likely have been pushed back onto the reef crests, and likely also westward along the southern rim of the atoll. However, in the interest of reducing the size of the manuscript, and given that the sewage outfalls are only one source of local nutrient loading (until recently, less than half of S Tarawa residents had toilets, i.e. lived in homes connected to sewage outfalls), we decided to remove this analysis from the manuscript.

RESULTS

- Lines 516-532 - I think that it would be a good idea to run these analyses with the two atolls separately along with the grouped analysis that you've already done. I think the version in the manuscript is informative, but I am curious about how presence/absence of certain taxa on each atoll affects the interpretation of these results. Breaking it up by atoll would also allow you to look at finer scale change over time. If you did do this, and I somehow missed it, it would be helpful to improve clarity about which analyses were done, as this was a bit hard to follow.

Thank you for making this point. We had attempted to run all of the analyses with the two atolls separately, but we were unable to for almost all of them because we did not have enough observations for some of the key taxa. We noted this in the original manuscript but understand that this information was most likely lost in the text because of the overwhelming number of analyses and the unclear structure of the paper. 

We did run an LMM specifically for massive Porites within Abaiang because we had large enough observation sizes and because we specifically wanted to test if the percent cover had changed for massive Porites because of the CoTs outbreak. The methods for this analysis are explained in lines 340 – 345, and the results are described in lines 555 – 562.

- Why did you decide to use annual mean SST versus maximum monthly mean or similar?

We only used the annual mean SST in one part of the analysis, when investigating whether SST influenced the percent cover of Halimeda (we use CVSST for the remainder of the analyses to capture the variation in temperatures over time). We had included this analysis because we wanted to investigate whether there was any relationship between ENSO dynamics and the percent cover of Halimeda (which influences the annual SST values; SSTs would be higher during El Niño events and lower during La Niña years). However, given the length of the manuscript and the complexity of the data we discuss, we have decided to remove this analysis from the manuscript.

- The results of the PERMANOVA as reported a little unconventionally. Specifically, in line 606 "The factor contributing most to the variation in the percent cover was 'year and mean NDVI'". I believe this should be reported as "the interaction between year and mean NDVI". That is, percent cover changes differently across years based on mean NDVI. This should be more clearly stated.

Thank you for this feedback. We have corrected this in the manuscript.

- I also think that the PERMANOVA should be run again, and separately for each island. I am not convinced that there isn't bias based on which locations were sampled during each field season, and would be convinced if you found similar results with individual models. I noticed that you ran the SIMPER for each atoll specifically, which I think was a good idea.

Thank you for this suggestion. We ran the PERMANOVA separately for each atoll and have updated the manuscript accordingly. As part of the reorganization of the manuscript, we have added subsections to the results to make them easier to follow (and to clearly relate them back to our hypotheses, please see our response to the first reviewer above). We have included the updated PERMANOVA results in the subsection titled “The Role of Human Disturbance, beginning on line 573. 

DISCUSSION

- The discussion is too long, and should be streamlined considerably. One thing that would be helpful would be careful thought toward the overall organization, and trimming unnecessary information that doesn't speak directly to your results and conclusions. For example, the paragraph from lines 755-769 is mostly unnecessary.

Thank you for this feedback. We have taken this into consideration and attempted to streamline the discussion, per your comments. We also removed/rewrote lines 755-769 as you suggested.

Specifically, we restructured the discussion to address the four hypotheses we describe in our response to Reviewer 1 above. We also added a section about the effects of the CoTs outbreak and how it affected the trajectories of the benthic communities within each atoll. We reduced the discussion to 4,626 words (from 5,230 words in the original manuscript). The revised discussion also includes new paragraphs to discuss the possibility of alternative stable states, per the suggestion from Reviewer 1 above.

- Lines 1037-1038 - "we therefore do not find it likely that the presence of Halimeda in Abaiang was triggered by any acute disturbances that allowed the macroalgae to outcompete corals." I do not think this is well supported by the results, as you don't know the history before the 1990s, and whether the reefs did (or did not) rapidly change from coral dominance to Halimeda dominance.

Thank you for this important point. We agree that we do not have enough evidence to back up this assertion and have removed it from the manuscript.

- When talking about the percent cover metrics in the context of calling a reef "healthy" or "degraded" I think the point (that percent coral doesn't tell you everything) is good (e.g. in lines 1038-1045), however, in the following paragraph (lines 1047-1061), you label those reefs dominated by P. rus to be "resilient". I think it would be worthwhile to consider, and explicitly define, what makes a reef resilient. If a reef with relatively high coral cover isn't "healthy" (e.g., sites at Tarawa), is it actually resilient? Sure, it is resilient as it seems like this may be a stable state that the ecosystem tends towards under these conditions, but under that definition, an algal dominated reef is also resilient, just toward a different state. Usually in this context resilient means something more - that reefs will maintain their structure and function - which is likely not true at Tarawa. I think this is an important point that needs to be discussed in the manuscript.

We agree with your important point that this discussion is warranted and that the manuscript is incomplete without it. Ironically, in our call for using more precise language when discussing the health and resilience of coral reefs, were imprecise in using the term ‘resilient’ where ‘resistant’ was more appropriate. We have added a discussion of the definition of ‘resilience’ (which includes but is not limited to resistance to disturbance, and whether sites in Tarawa meet that definition) per your suggestion and we agree that it was an important concept to address. 

Specifically, we added the following (beginning at line 886, in a section of the discussion titled “Disturbance, reef health, and resilience”): 

“However, for a coral reef to be considered resilient, it must be both resistant to disturbance and be able to recover to the original community structure post-disturbance, without an associated loss in function and services [90]. We have provided further evidence that the P. rus-dominated reefs in Tarawa are more resistant to heat stress than those in Abaiang, which is in agreement with previous studies [19]. Sites in Abaiang may become more resistant to heat stress in the future, depending on their recovery trajectory (if, for example, coral cover increases and is composed of more thermally tolerant genera). Also, because sites in Tarawa were dominated by a single coral species (P. rus), they are potentially vulnerable to future ‘ecological surprises’ [92]; any disturbance that has a disproportionate impact on P. rus could have a severe impact on the ecosystem as a whole. There was also likely a loss in function and services associated with the phase shift to P. rus at sites in Tarawa, which indicates that while these reefs are resistant to heat stress, they may not be resilient. Surveys of fish and invertebrate assemblages, as well as and reef structural complexity, would provide useful information about how the phase shift to P. rus is influencing the ecosystem services that are valuable to people residing in Tarawa.”

FIGURES

Figure 3 - Is it possible to include earlier dates for historical warming at this location?

Yes, we have updated the figure to include the full dataset (from 1985 – 2018) per your suggestion.

Figure 4 - Since you mention the difference between macroalgae and turf, would it be possible to include turf on both of these panels, in addition to live coral and macroalgae?

Thank you for this recommendation. We have redone the figure to include turf algae, in addition to live coral and macroalgae. We removed the dotted line for the corrected live coral values and instead used those corrected values to represent the live coral for Tarawa, because it was difficult to see the two separate lines representing live coral cover (one for the original values and one for the corrected values) after adding the line for turf algae. We discuss our reasoning for the corrected values in the text of the manuscript instead of showing both corrected and uncorrected values on the plot.

Figure 5 - Pocillopora is spelled incorrectly

Thank you for catching this error. We decided to remove the PCA from the manuscript because the information it includes is largely captured through the SIMPER analysis as well as the stacked bar chart.

EDITS

Line 20 - change "in" to "on"

Line 26 - change "The" to "These"

Line 27 - remove "the"

Lines 65-66 - I'm not convinced about saying "most commonly" here, and I'd suggest removing it

Line 122 - rephrase "more recent work questions that conclusion" to "this conclusion is still under debate"

Line 202 - add in "Hawaii" after Kane`ohe Bay

Line 217 - replace "continue" with "continued"

Line 219 - remove "researchers found that"

Line 220 - replace "had been" with "was"

Line 224 - replace "the" with "these"

Line 260 - does "ocean side" mean "reef slope" please be a little more specific

Line 375 - remove "the" from "For the percent cover"

Line 470 - Table 2 - what is "other morphology/species" for Porites?

Thank you for these suggestions. We have made all of the corrections in the manuscript. In Table 2, other morphologies/species of Porites was encrusting Porites. We have removed this table and instead described these numbers in the text of the manuscript, and we’ve changed ‘other morphology/species’ to ‘encrusting. By ‘ocean side’, we meant the outside rim of the atoll (in other words, not in the lagoon). We have clarified this in the manuscript.

References

1. Veron J. Corals of the World. Vols. 1–3. 2000. 

2. McClanahan T, Darling E, Maina J, Muthiga N, D’agata S, Leblond J, et al. Highly variable taxa-specific coral bleaching responses to thermal stresses. Mar Ecol Prog Ser. 2020 Aug 27;648:135–51. 

3. Donner SD, Kirata T, Vieux C. Recovery from the 2004 bleaching event in the Gilbert Islands, Kiribati. Atoll Research Bulletin. 2010;587:1–27. 

4. Carilli J, Donner SD, Hartmann AC. Historical temperature variability affects coral response to heat stress. PLoS ONE. 2012;7(3):1–9. 

5. Lovell E. Coral Reef Benthic Surveys of Tarawa and Abaiang Atolls, Republic of Kiribati. Tarawa, Kiribati: The South Pacific Applied Geoscience Commission; 2000 Aug p. 1–88. Report No.: 310.

---

## [Decision Letter · Decision Letter 1]

2 Jun 2021

PONE-D-21-04308R1

Coral reefs in the Gilbert Islands of Kiribati: Resistance, resilience, and recovery after more than a decade of multiple stressors

PLOS ONE

Dear Dr. Cannon,

Thank you for submitting your revised manuscript to PLOS ONE. After two further reviews, we would like to invite you to submit a revised version of the manuscript that addresses the points raised during the review process.

Both reviewers agreed that you had done a good job of addressing their first set of comments and both agreed that minor revisions are needed before acceptance. Reviewer 1 felt the manuscipt is still too long and requested that you summarise and reduce some more to focus on the key take home points of your paper. R1 felt some of the tables were unnecessary and could be provided as supporting info. Both reviewers asked that you don't use the acronym "ASS" for reasons that I hope are clear! 

Please submit your revised manuscript within the next 45 days. If you will need more time than this to complete your revisions, please reply to this message or contact the journal office at plosone@plos.org. Please include the following items when submitting your revised manuscript:

We look forward to receiving your revised manuscript.

Kind regards,

James R. Guest, Ph.D.

Academic Editor

PLOS ONE

Journal Requirements:

Reviewers' comments:

Reviewer's Responses to Questions

**Comments to the Author**

1. If the authors have adequately addressed your comments raised in a previous round of review and you feel that this manuscript is now acceptable for publication, you may indicate that here to bypass the “Comments to the Author” section, enter your conflict of interest statement in the “Confidential to Editor” section, and submit your "Accept" recommendation.

Reviewer #1: (No Response)

Reviewer #2: (No Response)

2. Is the manuscript technically sound, and do the data support the conclusions?

Reviewer #1: Yes

Reviewer #2: Yes

3. Has the statistical analysis been performed appropriately and rigorously? 

Reviewer #1: Yes

Reviewer #2: Yes

4. Have the authors made all data underlying the findings in their manuscript fully available?

Reviewer #1: Yes

Reviewer #2: No

5. Is the manuscript presented in an intelligible fashion and written in standard English?

Reviewer #1: Yes

Reviewer #2: Yes

6. Review Comments to the Author

Reviewer #1: Cannon et al. have made an adequate attempt to revise their manuscript, and this version is better than the first. Overall, the science message is fine and the results support the conclusion. The text remains parochial with limited appeal to readers who don’t work on coral reefs or even those who don’t have interest in the Gilbert Islands, and the manuscript remains far longer than it needs to be. This sort of material does not need 7 tables (including 2 full page versions) to summarize and present the pertinent data, and further consideration should be given to moving these to supplemental material, or replacing them with a more synthetic data summary. The “Future Directions” and “Conclusions” statements should be dropped from the manuscript since they either contain material that is not relevant to the questions/hypothesis guiding this work or are duplicative. Let’s not propagate the use of “ASS” as an acceptable acronym in coral reef literature.

Reviewer #2: Overall, I think that the authors did a good job of reorganizing the manuscript and clarifying their questions and hypotheses. I have just a few small comments on the revised version, but after they are addressed, I think this manuscript is ready for publication.

Edits

Line 23 - 'year' should be 'years'

Line 28 - repeated phrase

Line 113 - 'sewerage' should be 'sewage'?

Reference entry errors - lines 474, 483,

Line 688 (and following) - I think you should just use the whole phrase 'alternative stable state', as I just couldn't get past repeatedly reading 'ASS'.

Lines 762-766 - It feels a little weird to include these specific numbers without the associated error, especially since the errors overlap in at least some cases. I think you could leave it as is, but add in the errors to the numbers in parentheses; e.g., "(from 1.09% in 2014 to 2.90% in 2018)"

Lines 791-792 - I am a little hesitant about this, as it seems like you're using a lack of evidence to suggest that there aren't chronic disturbances in Abaiang. Although I know that Abaiang is much less populated than Tarawa, it still does have a human population, and so there could be disturbances that you aren't measuring. Please soften this statement to acknowledge that possibility.

Line 930 - repeated word

Line 935 - change 'what' to 'which'

7. PLOS authors have the option to publish the peer review history of their article (what does this mean?). If published, this will include your full peer review and any attached files.

Reviewer #1: No

Reviewer #2: No

---

## [Author Response · Author response to Decision Letter 1]

15 Jun 2021

15 June, 2021

Re: PONE-D-21-04308, “Coral reefs in the Gilbert Islands of Kiribati: Resistance, resilience, and recovery after more than a decade of multiple stressors”

Dear Dr. Guest,

Thank you for the detailed and helpful feedback and the two anonymous reviews you have shared with us for our recent manuscript submission to PLOS ONE. On behalf of my co-authors, I am submitting a revised manuscript that responds to the very helpful suggestions and concerns raised by the reviewers.

In response to the reviewer’s comments, we have removed the acronym ‘ASS’ (although please note that we did not invent this acronym; it was used in the literature we cited) and removed the first full table to the supplementary materials, have integrated the future directions section into the discussion (with less detail), and have added a paragraph to explain the importance of these results to coral reefs outside of the Gilbert Islands. 

I am including my responses to each of the comments below (the original comment is in regular text and my response is in italics). I have highlighted where we made changes in the manuscript to address the comments and have also uploaded two versions of the revised manuscript (one that tracks the changes, and one that does not), per your instruction.

Thank you again for your response and suggestions, and for the opportunity to respond. We look forward to hearing back from you.

Sincerely,

Sara E. Cannon, M.Sc., Ph.D. Candidate

University of British Columbia

 

Reviewer #1: Cannon et al. have made an adequate attempt to revise their manuscript, and this version is better than the first. Overall, the science message is fine and the results support the conclusion. The text remains parochial with limited appeal to readers who don’t work on coral reefs or even those who don’t have interest in the Gilbert Islands, and the manuscript remains far longer than it needs to be. This sort of material does not need 7 tables (including 2 full page versions) to summarize and present the pertinent data, and further consideration should be given to moving these to supplemental material, or replacing them with a more synthetic data summary. The “Future Directions” and “Conclusions” statements should be dropped from the manuscript since they either contain material that is not relevant to the questions/hypothesis guiding this work or are duplicative. Let’s not propagate the use of “ASS” as an acceptable acronym in coral reef literature.

Thank you for your helpful feedback. We have moved the first table with all of our site information to the supplementary materials to further reduce the number of tables in the manuscript and have added more information in the introduction and the discussion sections about why this work is important for coral reefs in other parts of the world (although we understand that it may not be relevant to people working outside of coral reef ecology).

Specifically, we edited the second paragraph of the abstract to read:

“The coral reefs of the Republic of Kiribati’s Gilbert Islands are exposed to frequent heat stress caused by central-Pacific type El Niño events, and may provide a glimpse into the future of coral reefs in other parts of the world, where the frequency of heat stress events will likely increase due to climate change.”

We also added a similar message to lines 45-50 of the abstract, which now read:

“These findings provide a rare glimpse at the future of coral reefs around the world and the ways they may be affected by climate change, which may allow scientists to better predict how other reefs will respond to increasing heat stress events across gradients of local human disturbance. We end with suggestions for future research that would address gaps in our analyses, with implications for coral reefs and the people depending on them, both locally and globally.”

In the introduction, we added at line 81:

“The coral reefs of Tarawa Atoll and its less populated neighbor Abaiang Atoll in the Republic of Kiribati provide a unique opportunity to investigate the role of chronic human disturbances on coral reef recovery from acute disturbances, and could also serve as examples of the ways that reefs in other parts of the world may respond to increasing frequencies of climate-driven heat stress events in the future.”

We also added, at line 184:

“Our findings provide a rare glimpse at how coral reefs around the world may respond to the increasing frequencies of heat stress events across gradients of local human disturbance and could provide important lessons to guide the future management of coral reef resources in the face of climate change.”

Finally, in the conclusion, we removed the sections “Future Directions” and “Conclusions” per your suggestion, and added a paragraph to summarize the importance of this work for reefs in other places in the future, beginning at line 956:

“Coral reefs in the Gilbert Islands have experienced years with prolonged heat stress more frequently than 99% of the world’s coral reefs [19], but this may change in the future; other reefs will likely experience more heat stress going forward, given that climate change-driven global coral bleaching events have increased in frequency and are expected to continue increasing in the due to climate change [93]. Reefs in the Gilbert Islands could therefore provide a rare glimpse into what reefs may look like in the future while also accounting for a gradient of local human impacts. We hope this work, coupled with further investigations into the potential trade-offs associated with locally degraded but climate-resistant reefs such as found in Tarawa versus reefs with more long-term ability to recover but with less resistance such as those in Abaiang, will provide novel information that is important for the future management of coral reef resources. Specifically, these findings may help to predict the ways that climate change will affect the millions of people around the world who depend on coral reefs, so that they may prepare for the future.”

We have also removed the acronym ASS per your suggestion and agree that it is less than ideal, although please note that it was not our invention (the acronym is already in use in several of the publications that we cited). 

 

Reviewer #2: Overall, I think that the authors did a good job of reorganizing the manuscript and clarifying their questions and hypotheses. I have just a few small comments on the revised version, but after they are addressed, I think this manuscript is ready for publication.

Edits

Line 23 - 'year' should be 'years'

Line 28 - repeated phrase

Line 113 - 'sewerage' should be 'sewage'?

Reference entry errors - lines 474, 483,

Line 688 (and following) - I think you should just use the whole phrase 'alternative stable state', as I just couldn't get past repeatedly reading 'ASS'.

Lines 762-766 - It feels a little weird to include these specific numbers without the associated error, especially since the errors overlap in at least some cases. I think you could leave it as is, but add in the errors to the numbers in parentheses; e.g., "(from 1.09% in 2014 to 2.90% in 2018)"

Thank you for these suggested edits. We have made the changes in the manuscript. 

Lines 791-792 - I am a little hesitant about this, as it seems like you're using a lack of evidence to suggest that there aren't chronic disturbances in Abaiang. Although I know that Abaiang is much less populated than Tarawa, it still does have a human population, and so there could be disturbances that you aren't measuring. Please soften this statement to acknowledge that possibility.

We appreciate you pointing this out and we agree that chronic disturbances likely do exist in Abaiang, although at much less intensity/severity than in Tarawa. We have updated the text to now read the following, on line 857-858: “We have no evidence that either of these two conditions currently exist in Abaiang.”

Line 930 - repeated word

Line 935 - change 'what' to 'which'

Thank you for your comments and suggested edits. We have removed this section from the manuscript per the suggestion of the other reviewer.

---

## [Editor Report · Decision Letter 2]

14 Jul 2021

Coral reefs in the Gilbert Islands of Kiribati: Resistance, resilience, and recovery after more than a decade of multiple stressors

PONE-D-21-04308R2

Dear Dr. Cannon,

We’re pleased to inform you that your manuscript has been judged scientifically suitable for publication and will be formally accepted for publication once it meets all outstanding technical requirements.

Kind regards,

James R. Guest, Ph.D.

Academic Editor

PLOS ONE

Additional Editor Comments (optional): I think you've dealt well with all of the suggested changes. I still feel that the paper is probably longer than it needs to be, but that in itself is not a reason not to accept it. I have one minor suggestion which is to change coral "length" to coral "diameter" in the methods as this is a more commonly used term.

---

## [Editor Report · Acceptance letter]

21 Jul 2021

PONE-D-21-04308R2 

Coral reefs in the Gilbert Islands of Kiribati: resistance, resilience, and recovery after more than a decade of multiple stressors 

Dear Dr. Cannon:

I'm pleased to inform you that your manuscript has been deemed suitable for publication in PLOS ONE. Congratulations! Your manuscript is now with our production department. 

Kind regards, 

on behalf of

Dr. James R. Guest 

Academic Editor

PLOS ONE